# A conserved cell-pole determinant organizes proper polar flagellum formation

Erick E Arroyo-Pérez[1,2]*[†], John C Hook[3†], Alejandra Alvarado[4], Stephan Wimmi[1,5], Timo Glatter[6], Kai Thormann[3]*, Simon Ringgaard[1,2]

[1]Max Planck Institute for Terrestrial Microbiology, Department of Ecophysiology, Munich, Germany; [2]Department of Biology I, Microbiology, Ludwig-Maximilians-Universität München, Munich, Germany; [3]Department of Microbiology and Molecular Biology, Justus-Liebig-Universität Giessen, Giessen, Germany; [4]Interfaculty Institute of Microbiology and Infection Medicine Tübingen, Bacterial Metabolomics, University of Tübingen, Tübingen, Germany; [5]Institute for Biological Physics, University of Cologne, Köln, Germany; [6]Core Facility for Mass Spectrometry and Proteomics, Max Planck Institute for Terrestrial Microbiology, Marburg, Germany

**\*For correspondence:**
erick.arroyo.perez@umontreal.
ca (EEA-P);
Kai.Thormann@mikro.bio.uni-
giessen.de (KT)

[†]These authors contributed
equally to this work

**Competing interest:** The authors
declare that no competing
interests exist.

**Reviewing Editor:** Karine A
Gibbs, University of California,
Berkeley, United States

## eLife assessment

This **important** study describes the discovery of a mechanism by which multiple species of bacteria synthesize and localize polar flagella via a novel protein, FipA, which interacts with FlhF. The authors use appropriate methodological approaches (biochemistry, molecular microbiology, quantitative microscopy, and bacterial genetics) to obtain and present **convincing** results and interpretations. This work will particularly interest those studying bacterial motility and bacterial cell biologists.

**Abstract** The coordination of cell cycle progression and flagellar synthesis is a complex process in motile bacteria. In γ-proteobacteria, the localization of the flagellum to the cell pole is mediated by the SRP-type GTPase FlhF. However, the mechanism of action of FlhF, and its relationship with the cell pole landmark protein HubP remain unclear. In this study, we discovered a novel protein called FipA that is required for normal FlhF activity and function in polar flagellar synthesis. We demonstrated that membrane-localized FipA interacts with FlhF and is required for normal flagellar synthesis in *Vibrio parahaemolyticus*, *Pseudomonas putida*, and *Shewanella putrefaciens*, and it does so independently of the polar localization mediated by HubP. FipA exhibits a dynamic localization pattern and is present at the designated pole before flagellar synthesis begins, suggesting its role in licensing flagellar formation. This discovery provides insight into a new pathway for regulating flagellum synthesis and coordinating cellular organization in bacteria that rely on polar flagellation and FlhF-dependent localization.

## Introduction

Many cellular processes depend on a specific spatiotemporal organization of its components. The DNA replication machinery, cell division proteins, and motility structures are some examples of elements that need to be positioned at a particular site during the cell cycle in a coordinated manner to ensure adequate cell division or proper function of the structures. To carry out this task, many cellular components of bacteria have a polar organization, that is they are asymmetrically

positioned inside the cell, which is particularly apparent in monopolarly flagellated bacteria. In the marine bacteria *Vibrio*, which constitutively express a single polar flagellum (*McCarter, 1995*; *Kim and McCarter, 2000*), it is crucial to coordinate the localization and timing of flagellar components, in order to guarantee that newly born cells only produce a flagellum in timing with cell division. This process is coordinated with the chromosome replication and the chemotaxis clusters through a supramolecular hub that is tethered at the old cell pole (*Takekawa et al., 2016*; *Yamaichi et al., 2012*). This organization depends on ATPases of the ParA/MinD family, which regulate the migration of the hub components to the new cell pole as the cell cycle progresses. In this way, the new cell has a copy of the chromosome and a set of chemotaxis clusters positioned next to what will be the site for the new flagellum. Central to this process is the protein HubP, which recruits to the pole the three ATPases responsible for the localization of these organelles: ParA1 for the chromosome (*Fogel and Waldor, 2006*), ParC for the chemotaxis clusters (*Ringgaard et al., 2011*) and FlhG for the flagellum (*Correa et al., 2005*; *Arroyo-Pérez and Ringgaard, 2021*). Homologs of HubP occur in several species, for example *Pseudomonas* (where it is called FimV), *Shewanella,* or *Legionella*, where they similarly act as organizers (hubs) of the cell pole (*Wehbi et al., 2011*; *Rossmann et al., 2015*; *Coil and Anné, 2010*).

The bacterial flagellum is a highly intricate and complex subcellular structure. It is composed of around 25 different proteins, which have to be assembled in different stoichiometries in a spatiotemporally coordinated manner (*Macnab, 2003*; *Chevance and Hughes, 2008*). Although there may be significant differences among species, flagella in general are composed of a motor attached to the cytoplasmic membrane, a rod connecting it to the extracellular space, a hook and a filament. The basal body is composed of a cytoplasmic C-ring or switch complex (*Francis et al., 1994*), which receives the signal from the chemotaxis proteins via binding to phosphorylated CheY. The C-ring is connected to the MS-ring, which is embedded in the cytoplasmic membrane (*Homma et al., 1987*; *Ueno et al., 1992*). Attached to it, on the cytoplasmic side, is the export apparatus, which allows secretion of rod, hook, and filament components (*Minamino, 2014*). Flagella can also be sheathed, if an extension of the outer membrane covers the filament, as is the case in many *Vibrio* species (*Chen et al., 2017*).

The positioning of the flagella and control of flagellar numbers per cell are mediated in many bacteria by the interplay between two flagellar regulators, FlhF and FlhG. Studies in a wide variety of proteobacteria indicate that FlhG is a negative regulator that restricts the number of flagella that are synthesized. In organisms such as strains of *Vibrio*, *Shewanella putrefaciens* and *Pseudomonas aeruginosa*, the absence of MinD-like ATPase FlhG results in hyper-flagellated cells (*Kusumoto et al., 2006*; *Arroyo-Pérez and Ringgaard, 2021*; *Schuhmacher et al., 2015a*; *Blagotinsek et al., 2020*; *Campos-García et al., 2000*; *Murray and Kazmierczak, 2006*). In contrast, in polarly flagellated species such as *Campylobacter jejuni, Vibrio cholerae, Vibrio parahaemolyticus, P. aeruginosa, S. putrefaciens* and *Shewanella oneidensis*, the SRP-type GTPase FlhF is a positive regulator of flagellum synthesis and necessary for proper localization. A deletion of *flhF* results in reduced number and mis-localization of flagella (*Hendrixson and DiRita, 2003*; *Correa et al., 2005*; *Arroyo-Pérez and Ringgaard, 2021*; *Pandza et al., 2000*; *Rossmann et al., 2015*; *Gao et al., 2015*; *Navarrete et al., 2019*; *Zhang et al., 2020*).

The current model predicts that GTP-bound dimeric FlhF (*Bange et al., 2007*; *Kondo et al., 2018*) localizes to the cell pole to where it recruits the initial flagellar building blocks (*Green et al., 2009*). A long-standing question is how FlhF recognizes and localizes to the designated cell pole. In *P. aeruginosa* and *V. alginolyticus,* it was demonstrated that FlhF localized polarly upon ectopic production in the absence of the flagellar master regulator FlrA (*Green et al., 2009*; *Kondo et al., 2017*). It was therefore speculated that FlhF assumes its polar localization in the absence of other flagellar proteins or even without any further additional protein factors. The underlying mechanism, however, remains enigmatic, given that neither in its monomeric nor its dimeric form, FlhF possesses any regions that would indicate a membrane association. In a recent study, it was shown that in the polarly flagellated gammaproteobacterium *S. putrefaciens* FlhF binds the C-ring protein FliG via a specific region at the very N-terminus of FlhF. The FlhF-FliG complex is then recruited to the designated cell pole by HubP, where FlhF-bound FliG captures the transmembrane protein FliF and promotes formation of the MS-ring. This forms the scaffold from where further flagellar synthesis can occur (*Dornes et al., 2024*). However, in the absence of HubP, the majority of cells still forms normal polar flagella, indicating that HubP is not the only polarity factor in this process (*Rossmann et al., 2015*).

Here, we provide evidence demonstrating that FlhF does not localize independently. Instead, we show that FipA, a small integral membrane protein with a domain of unknown function, facilitates the recruitment of FlhF to the membrane at the cell pole and, at least in some species, it acts in concert with the polar landmark protein HubP/FimV. Using *V. parahaemolyticus* and two additional polarly flagellated γ-proteobacteria, the monopolarly flagellated *S. putrefaciens* and the lophotrichous *Pseudomonas putida*, we show that FipA universally mediates recruitment of FlhF to the designated cell pole. The spatiotemporal localization behavior of FipA as well as its relationship to FlhF, support its role as a licensing protein that enables flagellum synthesis.

## Results
### Identification of an FlhF protein interaction partner: FipA

In order to identify potential factors required for FlhF function and its recruitment to the cell pole, we performed affinity purification of a superfolder green fluorescent protein (sfGFP)-tagged FlhF (FlhF-sfGFP) ectopically expressed in wild-type *V. parahaemolyticus* cells followed by shotgun proteomics using liquid chromatography-tandem mass spectrometry analysis (LC-MS/MS). Among the proteins that were significantly enriched in FlhF-sfGFP purifications, eight were structural components of the flagellum (*Figure 1A*; *Supplementary file 1a*), but also a non-flagellar protein, VP2224, was significantly co-purified with FlhF (*Figure 1A*; *Supplementary file 1a*). The homologue of VP2224 in *V. cholerae*, there named FlrD, was previously shown to exert the same activating role on flagellar gene regulation as FlhF (*Moisi et al., 2009*), suggesting a function related to FlhF. Notably, FlhF was also significantly co-purified in the reciprocal co-IP-MS/MS experiment using VP2224-sfGFP as bait (*Figure 1B*), suggesting a direct or indirect interaction of VP2224 and FlhF. To further investigate the co-purification data, bacterial two-hybrid (BACTH) assays in the heterologous host *E. coli* were carried out and suggested a direct interaction between FlhF and VP2224. FlhF and VP2224 were also found to self-interact (*Figure 1E*). Additionally, these data indicated that FlhF and VP2224 form an interaction complex in both the native and a heterologous host organism. Thus, we identified VP2224 as a novel interaction partner of FlhF and named it FipA for Flh*F* *i*nteraction *p*artner *A*.

### FipA constitutes a new family of FlhF interaction partners

The gene encoding FipA is located immediately downstream of the flagellar operon that encodes FlhA, FlhF, FlhG, FliA and the chemotaxis proteins (*Figure 1C*). In *V. parahaemolyticus*, FipA consists of 163 amino acids with a molecular mass of 18.4 kDa. In silico analysis and membrane topology mapping predicted that FipA consists of a short periplasmic N-terminal part, consisting of amino acids 1–5, followed by a transmembrane region (6-28), a cytoplasmic part harboring a coiled region (amino acids 31–58) and a domain of unknown function, DUF2802, positioned in the C-terminal half of the protein (from amino acids 68–135; *Figure 1D*; *Figure 1—figure supplement 1*).

InterPro database analyzes found that FipA homologues (i.e. membrane proteins consisting of a single cytoplasmic DUF2802 repeat) are widespread among γ-proteobacteria. Exceptions are the *Enterobacteriaceae*, which do not possess any copies of either *fipA* nor of *flhF* and *flhG*. Actually, FipA is only present in genomes that also encode FlhF and FlhG (*Figure 1F*), which prompted the question of whether FipA is involved in regulating the flagellation pattern in concert with the FlhF-FlhG system. By including in our analysis the flagellation pattern reported in the literature, we found that the species that encode FipA are all polar flagellates, either monotrichous or lophotrichous (*Figure 1F*). FipA homologues are absent from bacteria that use the FlhF-FlhG system to produce different flagellation patterns, like the peritrichous *Bacillus*, the amphitrichous ε-Proteobacteria or Spirochetes (*Supplementary file 1b*). In the α-Proteobacteria, where FlhF homologues are only present in a few species, FipA is absent as well (*Figure 1F*). Based on these analyzes, we hypothesized that FipA represents a new family of FlhF interaction partners important for the γ-proteobacteria. To test this hypothesis, we decided to analyze FipA-FlhF interaction in addition in the distantly related and lophotrichously flagellated *P. putida* and the monotrichous *S. putrefaciens*. BACTH analysis showed that the FipA orthologue from both *P. putida* (*Pp*FipA, PP_4331) and *S. putrefaciens* (*Sp*FipA, SputCN32_2550) interact with FlhF of their respective species and that they self-interact (*Figure 1—figure supplement 2*) – a result similar to that of FipA from *V. parahaemolyticus* (*Vp*FipA). This supported our hypothesis

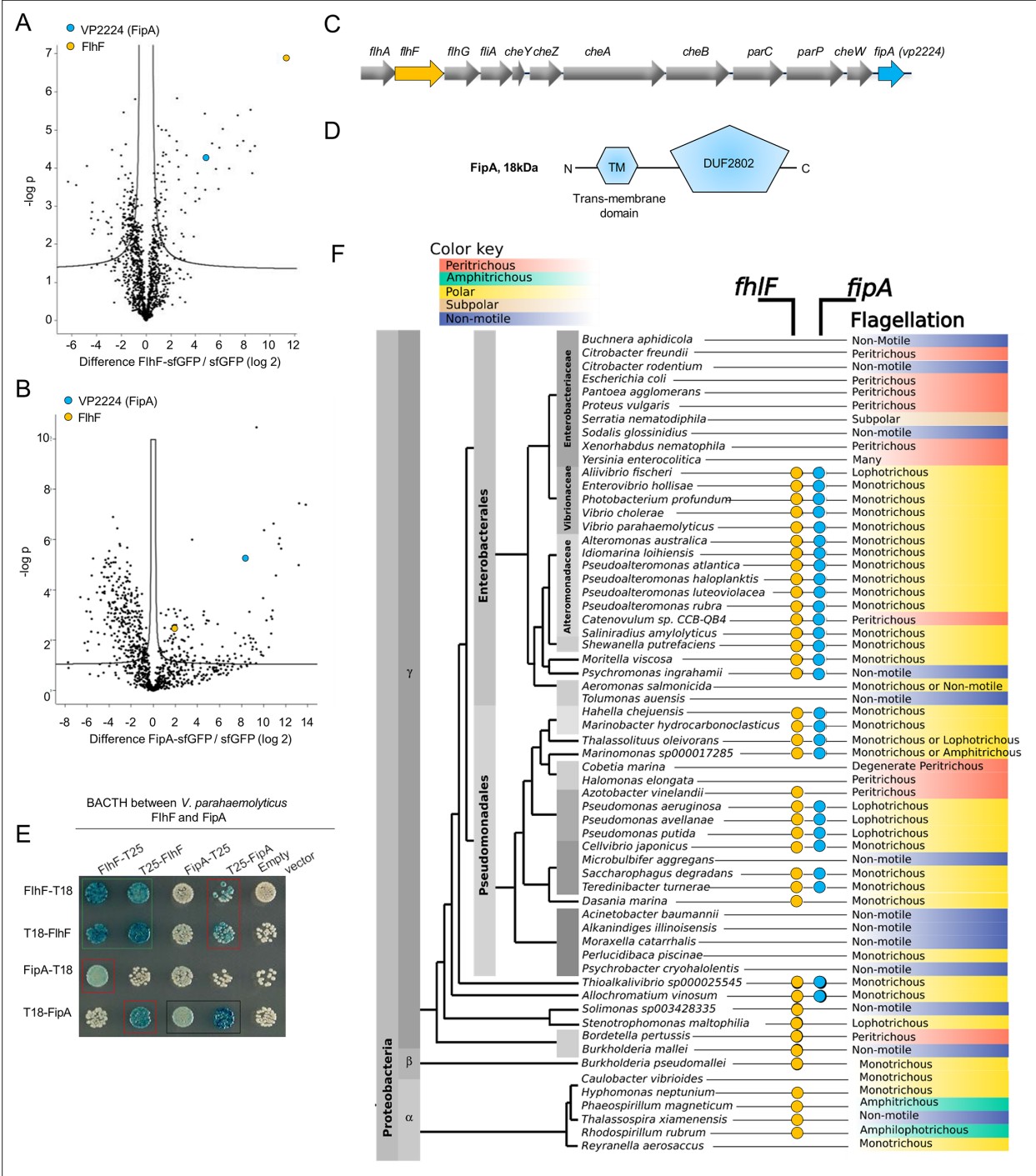

**Figure 1.** FipA constitutes a new family of FlhF interaction partners. Volcanoplots representing Log-ratios versus significance values of proteins enriched in (**A**) FlhF-sfGFP or (**B**) FipA-sfGFP purifications using shotgun proteomics and liquid chromatography-mass spectrometry; sfGFP was used as control. The full list of pulled-down proteins can be found in the ***Supplementary file 1a***. (**C**) Organization of the flagellar/chemotaxis gene region encoding FlhF and FipA in *V. parahaemolyticus*. (**D**) Domain organization of FipA (TM, transmembrane region; DUF, domain of unknown function). (**E**) Bacterial two-hybrid confirming the interaction between FipA and FlhF from *V. parahaemolyticus*. The indicated proteins (FipA, FlhF) were fused N- or C-terminally to the T18- or T25-fragment of the *Bordetella pertussis* adenylate cyclase. In vivo interaction of the fusion proteins in *Escherichia coli* is indicated by blue color. The corresponding assay in *P. putida* and *S. putrefaciens* is displayed in ***Figure 1—figure supplement 2***. (**F**) Dendrogram of γ-proteobacteria, indicating the presence of FlhF or FipA homologues and the corresponding flagellation pattern. An extended version and sources are available in ***Supplementary file 1b***.

The online version of this article includes the following figure supplement(s) for figure 1:

*Figure 1 continued*

**Figure supplement 1.** Membrane topology of FipA.

**Figure supplement 2.** FlhF interacts with FipA in a bacterial two-hybrid analysis (BACTH) in *P. putida* and *S. putrefaciens*.

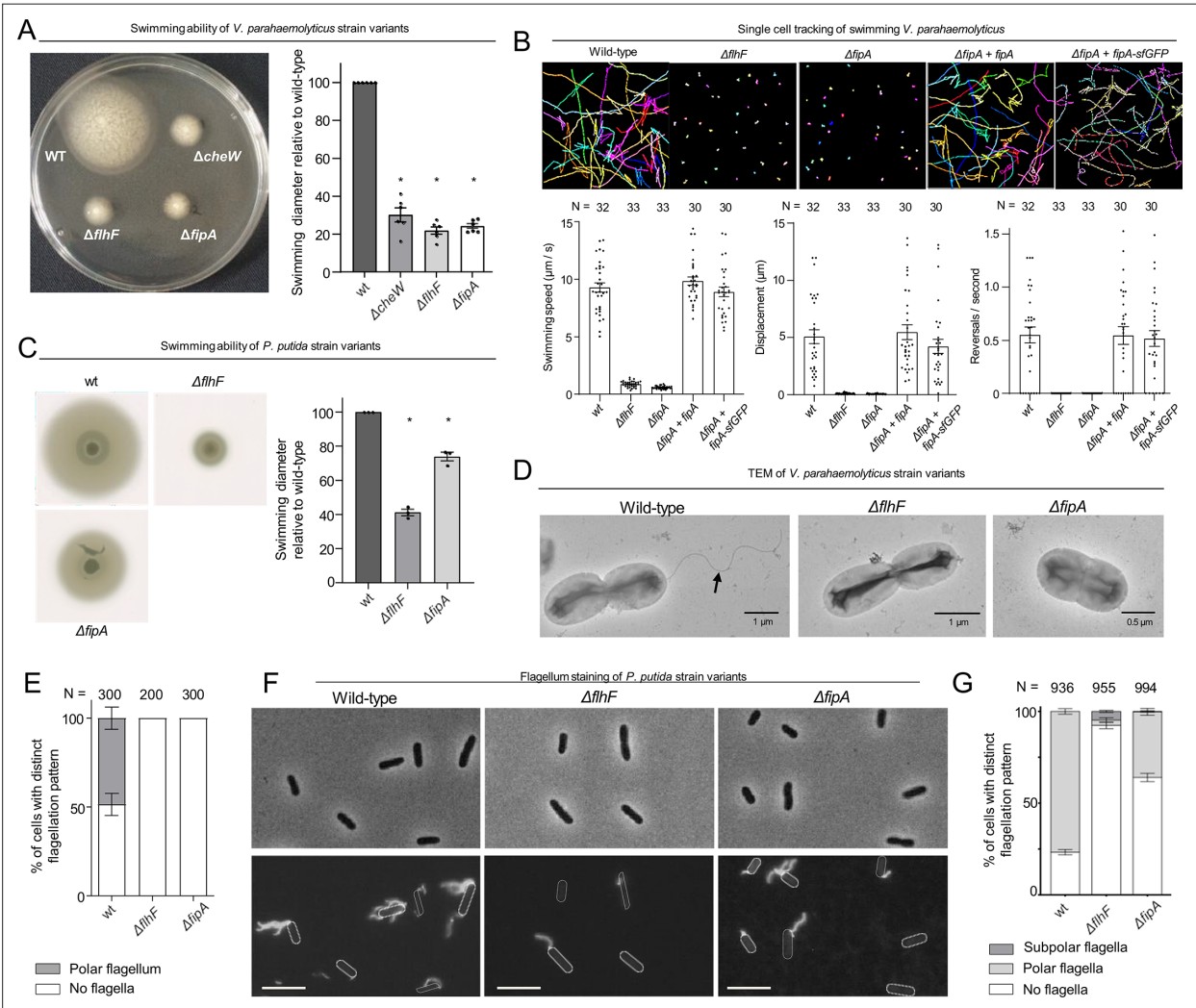

**Figure 2.** FipA is required for correct flagellum formation. (**A, C**) Representative soft-agar swimming assay of *V. parahaemolyticus* (**A**) or *P. putida* (**C**) strains (left panels) and the corresponding quantification (right panels). For the latter, the halo diameter measurements were normalized to the halo of the wild type on each plate. Data presented are from six (**A**) or three (**C**) independent replicates, asterisks represent a p-value < 0.05 (according to ANOVA + Tukey tests). (**B**) Single-cell tracking of *V. parahaemolyticus*. Shown are representative swimming trajectories and quantification of swimming speed, total displacement and reversal rate. N indicates number of cells tracked among three biological replicates (ANOVA + Tukey test). (**D**) Representative electron micrographs of the indicated *V. parahaemolyticus* strains stained with uranyl acetate. (**E**) Quantification of flagellation pattern in the populations of the indicated *V. parahaemolyticus* strains. (**F**) Flagellum stain of indicated *P. putida* strains with Alexa Fluor 488-C5-maleimide and (**G**) quantification of the corresponding flagellation in the population. N indicates the number of cells counted among three biological replicates. For *S. putrefaciens*, see **Figure 2—figure supplement 1**.

The online version of this article includes the following figure supplement(s) for figure 2:

**Figure supplement 1.** Deletions of or in FipA and FlhF affect *S. putrefaciens* flagellation.

and indicated that FipA has a general function as an FlhF interaction partner, thus constituting a new class of FlhF interaction partners.

## FipA is required for proper swimming motility and flagellum formation

To explore the role of FipA with respect to flagellation, we generated mutant strains with individual deletions of the *fipA* and *flhF* genes. Strikingly, absence of FipA in *V. parahaemolyticus* completely abolished swimming motility in soft-agar medium to the same degree as cells lacking FlhF (*Figure 2A*). Furthermore, single-cell tracking of planktonic *V. parahaemolyticus* cells confirmed that cells lacking FipA were completely non-motile and behaved identical to cells lacking FlhF (*Figure 2B*). Importantly, ectopic expression of FipA in the *ΔfipA* strain restored the strain's swimming ability to wild-type levels (*Figure 2B*), further supporting that it is the deletion of *fipA* that results in the phenotype and not polar effects resulting from the *fipA* deletion. The C-terminal FipA-sfGFP fusion used throughout this paper also restored the phenotype (*Figure 2B*), indicating that the fusion protein is fully functional.

The swimming phenotypes in the absence of FipA could result from defects in either flagellum assembly or in the flagellar motor performance. To differentiate between these possibilities, we examined *V. parahaemolyticus* by transmission electron microscopy (TEM; *Figure 2D and E*). Planktonic cells lacking either FlhF or FipA, showed a complete absence of flagella on the bacterial surface, while a single polar flagellum was observed in ~50% of wild-type cells (*Figure 2D and E*).

Based on this set of experiments, we concluded that FipA and FlhF are essential for normal formation of polar flagella.

## FipA and HubP ensure proper localization of FlhF to the cell pole

As our previous results showed an interaction between FipA and FlhF, we analyzed if the intracellular localization of FlhF was influenced by FipA. In this regard, we used functional translational fusions of FlhF to fluorescent proteins expressed from its native site on the chromosome in *V. parahaemolyticus* (*Figure 3—figure supplements 1 and 2*; *Arroyo-Pérez and Ringgaard, 2021*).

We observed that FlhF localized either diffusely in the cytoplasm or to the cell pole as previously reported (*Arroyo-Pérez and Ringgaard, 2021*). However, a significant delocalization of FlhF from the cell pole occurred in the absence of FipA (*Figure 3A–C*; *Figure 3—figure supplement 2*). Particularly, FlhF was diffusely localized in ~37% of cells or localized to the cell pole in a uni- and bi-polar manner in ~45% and~19%, respectively, compared to wild-type cells (*Figure 3B*). Absence of FipA significantly reduced localization of FlhF to the cell pole with a concomitant increase in diffusely localized FlhF (70%; *Figure 3A–C*; *Figure 3—figure supplement 2*). Furthermore, the foci of FlhF at the cell pole in the absence of FipA were significantly dimmer compared to wild-type FlhF foci (*Figure 3D*), indicating that the amount of FlhF localized to the cell pole is lower in the absence of FipA. Importantly, even though FlhF was still able to localize to the cell pole in a certain number of cells lacking FipA, no flagellum was formed in this background (*Figure 2D–E*), thus indicating that FipA not only is required for proper polar recruitment of FlhF, but also for the stimulation of flagellum formation.

Given the role of HubP in cell pole organization and its reported interaction with FlhF (*Yamaichi et al., 2012*), we also analyzed the localization of FlhF in the absence of HubP. As for FipA mutants, recruitment of FlhF to the cell pole was reduced in the absence of HubP. Strikingly, in the double deletion strain *ΔfipA ΔhubP*, FlhF did not localize as foci at the cell pole at all but was instead localized diffusely in the cytoplasm in 100% of cells (*Figure 3A–C*; *Figure 3—figure supplement 3*). Immunoblot analysis showed that the difference in localization of FlhF-sfGFP was not due to differences in expression levels or protein stability (*Figure 3—figure supplement 1*).

Altogether, these data suggest that both FipA and HubP work together to promote normal polar localization of FlhF.

## The transmembrane and conserved cytoplasmic domains of FipA is required for its function in regulating flagellum formation and proper FlhF localization

In order to identify the regions of FipA mediating its role in flagellum regulation, we next analyzed the N-terminal transmembrane domain for FipA membrane anchoring and its function in mediating proper flagellum production. When expressed in *E. coli*, a C-terminally GFP-tagged version of FipA (*Vp*FipA-sfGFP) showed a clear membrane localization, while a FipA variant deleted for the predicted

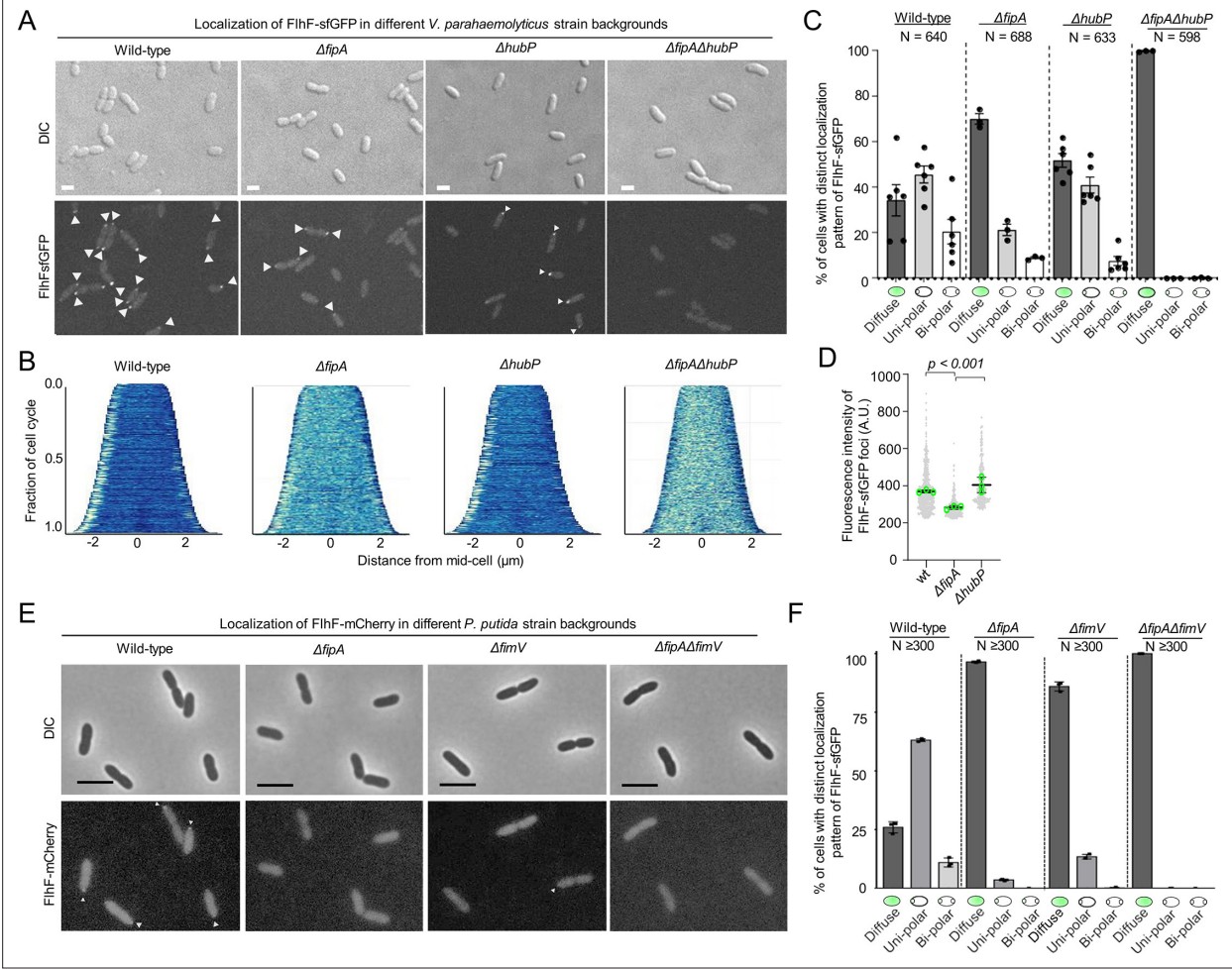

**Figure 3.** Localization of FlhF depends on FipA and HubP. (**A**) Representative micrographs of the indicated strains of *V. parahaemoloyticus* expressing FlhF-sfGFP from its native promoter. Upper panel shows the DIC image, the lower panel the corresponding fluorescence image. Fluorescent foci are highlighted by white arrows. Scale bar = 2 μm. For an enlargment see *Figure 3—figure supplement 3*. (**B**) Demographs displaying FlhF-sfGFP fluorescence intensity along the cell length within the experiments shown in (**A**). (**C**) Quantification of localization patterns and (**D**) foci fluorescence intensity of the fluorescence microscopy experiment presented in (**A**). The data was combined from the given number (**N**) of cells combined from three biological replicates. (**E**) Representative micrographs of the indicated *P. putida* strains expressing FlhF-sfGFP from its native promoter. The upper panels show the DIC and the lower panels the corresponding fluorescence images. Fluorescent foci are marked by small white arrows. The scale bar equals 5 μm. The low intensity of the foci did not allow a quantitative analysis of foci intensities or the generation of demographs. For an enlargement of the micrographs see *Figure 3—figure supplement 4*. (**F**) Quantification of FlhF-sfGFP localization patterns in the corresponding strains of *P. putida* from the experiments shown in (**E**). Corresponding data on the localization of FlhF in *S. putrefaciens* is displayed in *Figure 3—figure supplement 5*.

The online version of this article includes the following source data and figure supplement(s) for figure 3:

**Figure supplement 1.** Western analysis on the protein levels of FlhF and FipA fusions to fluorescent proteins.

**Figure supplement 1—source data 1.** Scans of the original western blots for *Figure 3—figure supplement 1*.

**Figure supplement 1—source data 2.** Scans of the original western blots for *Figure 3—figure supplement 1* with labels.

**Figure supplement 2.** N-terminal tagging of *Pp*FlhF with mCherry does not negatively affect spreading motility in soft agar.

**Figure supplement 3.** Localization of FlhF depends on FipA and HubP.

**Figure supplement 4.** Localization of FlhF depends on FipA and HubP.

**Figure supplement 5.** *Sp*FipA and *Sp*HubP concertedly affect polar *Sp*FlhF localization.

transmembrane (TM) domain between residues 7–27 (FipAΔTM), was diffusely localized in the cytoplasm (*Figure 4A*). These observations show that FipA is indeed a membrane protein anchored by the predicted transmembrane domain. To then study the function of FipA membrane localization, the native *fipA* locus was replaced by a gene encoding a FipA variant lacking the N-terminal TM domain,

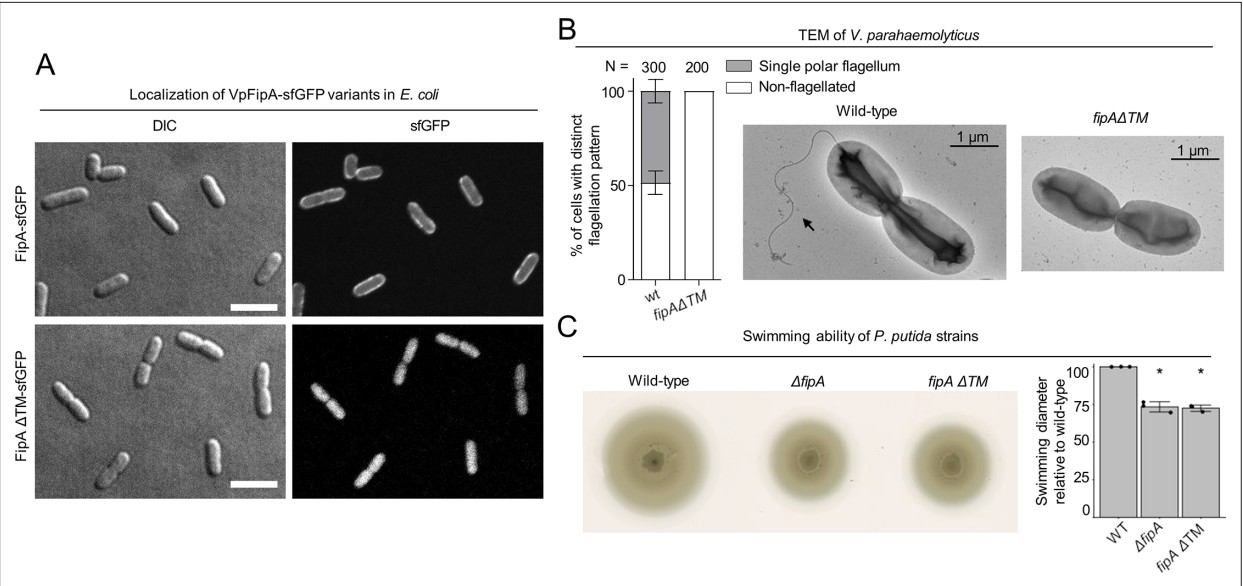

**Figure 4.** Activity of FipA depends on FlhF and on its transmembrane domain. (**A**) Micrographs of *E. coli* cells expressing FipA-sfGFP from *V. parahaemolyticus*, and a truncated version lacking the transmembrane domain (ΔTM). The left panels display the DIC and the right panels the corresponding fluorescence images. The scale bar equals 5 μm. (**B**) Electron micrographs of *V. parahaemolyticus* wild-type and mutant cells lacking the transmembrane domain of *fipA*, respectively. The corresponding quantification of the flagellation pattern is shown to the left of the micrographs. Note that the data for the wild-type cells is the same as in *Figure 2D and E*. (**C**) Spreading behavior of the indicated *P. putida* strains (left) with the corresponding quantification (right). Loss of the FipA TM region phenocopies a complete *fipA* deletion.

and the resulting strain (*fipA ΔTM*) was analyzed for flagellum production. The *V. parahaemolyticus* strain carrying this mutation did not produce any flagella at all (*Figure 4B*), as did the Δ*fipA* strain (*Figure 2D*).

After validating an essential role for membrane anchoring of FipA, we analyzed in more detail the cytoplasmic DUF2802 domain. By aligning various FipA homologs, we identified a motif of conserved amino acids, and three of them (G110, E126 and L129 of *Vp*FipA) were 100% conserved among FipA homologues (*Figure 5A*). Therefore, we chose these amino acids for mutagenesis. Alanine substitutions of residues G110 and L129 abolished the interaction between *Vp*FipA and *Vp*FlhF in BACTH analysis, while the E126A substitution did not affect the interaction (*Figure 5B*). All these variants were, however, still able to self-interact with wild-type *Vp*FipA (*Figure 5B*). Altogether, these results suggest that conserved residues in the DUF2802 domain support the interaction between FipA and FlhF, and that the self-interaction is mediated by a different region.

## Interaction between FipA and FlhF is required for FipA function on regulating flagellum formation and FlhF localization

We proceeded to evaluate the role of these residues in vivo. After introducing the mutations in the *fipA* gene in its native loci, the effects on motility were almost indistinguishable from a Δ*fipA* mutation. In *V. parahaemolyticus,* this resulted in completely non-motile cells in planktonic cultures (*Figure 5D*).

The localization of FlhF was also affected by the residue substitutions in *fipA*. Cells of *V. parahaemolyticus* expressing FlhF-sfGFP natively had reduced polar localization when FipA was substituted with either G110A or L129A (*Figure 5F*; *Figure 5—figure supplement 1*), with almost 50% of cells presenting only diffuse FlhF-sfGFP signal. Thus, it seems that the DUF2802 domain of FipA is responsible for the effect of FipA on motility, and that its effects occur primarily through the interaction with FlhF. Furthermore, the interruption of FipA-FlhF impedes the recruitment of FlhF to the cell pole.

## Cell cycle-dependent polar localization of FipA

Given FipA's function in regulating correct recruitment of FlhF to the cell pole, we next analyzed the intracellular localization of FipA. To this end, a hybrid gene bearing a fusion of *fipA* to *sfgfp* was used to replace native *fipA*, resulting in stable and functional production of FipA C-terminally

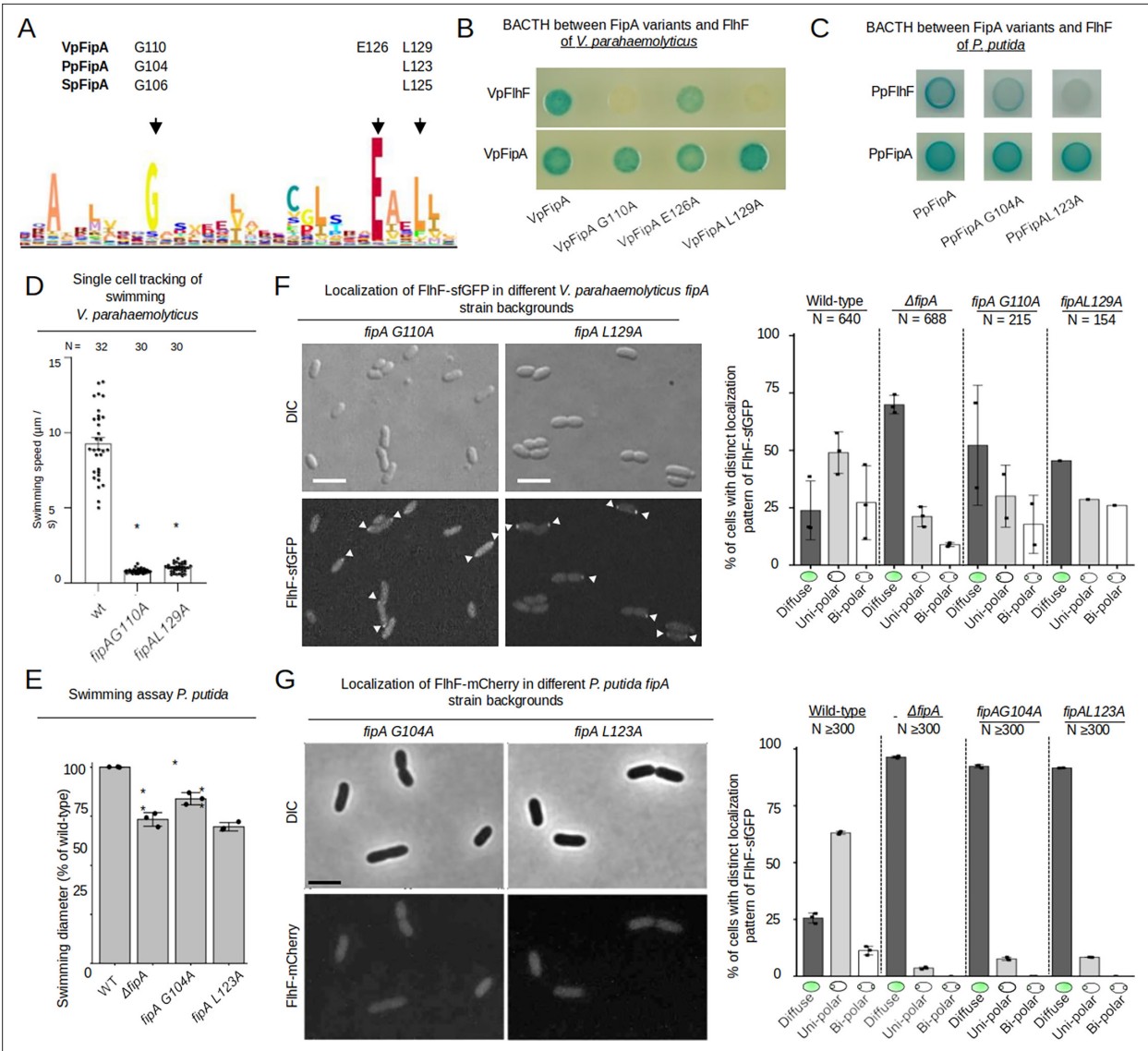

**Figure 5.** Conserved residues in the domain of unknown function of FipA are essential for interaction with FlhF. (**A**) Weight-based consensus sequence of the conserved region of DUF2802 as obtained from 481 species. The residues targeted in the FipA orthologs of *V. parahaemolyticus*, *P. putida* and *S. putrefaciens* are indicated along with their appropriate residue position. (**B, C**) Bacterial two-hybrid assay of FipA variants of *V. parahaemolyticus* (**B**) or *P. putida* (**C**) with an alanine substitution in the conserved residues indicated in (**A**). The constructs were tested for self-interaction and interaction with FlhF. In vivo interaction of the fusions in *E. coli* is indicated by blue coloration of the colonies. (**D**) Quantification from single-cell tracking of swimming *V. parahaemolyticus* cells expressing FipA bearing the indicated substitution in the DUF2802 domain (see *Figure 2B* for wild-type behavior). Asterisks indicate a p-value <0.05 (ANOVA + Tukey test) (**F**) Localization of *Vp*FlhF in the absence of FipA or in cells with substitutions in the DUF2802 domain. Left: Micrographs showing the localization of FipA-sfGFP in the indicated strains; the upper panels display the DIC and the lower panels the corresponding fluorescence images (for an enlargement see *Figure 5—figure supplement 1*). The scale bar equals 5 µm. Right: the corresponding quantification of the FlhF-sfGFP patterns in the indicated strains. (**E**) Soft-agar spreading assays of *P. putida* wild-type and indicated mutant strains, asterisks display a p-value of 0.05 (*) or 0.01 (**) (ANOVA). (**G**) Localization of *P. putida* FlhF in strains bearing substitutions in the DUF2802 interaction site. Left: micrographs displaying the localization of FlhF-mCherry in the indicated strains. Upper panels show the DIC and lower panels the corresponding fluorescence images. The scale bar equals 5 µm (for an enlargement see *Figure 5—figure supplement 1*). Right: Corresponding quantification of the FlhF localization pattern in the indicated strains. Data for *S. putrefaciens* is displayed in *Figure 5—figure supplement 2*.

The online version of this article includes the following figure supplement(s) for figure 5:

**Figure supplement 1.** Conserved residues in the domain of unknown function of FipA are essential for interaction with FlhF in *P. putida*.

**Figure supplement 2.** Targeted residue substitution in FipA affects FlhF positioning and function in *S. putrefaciens*.

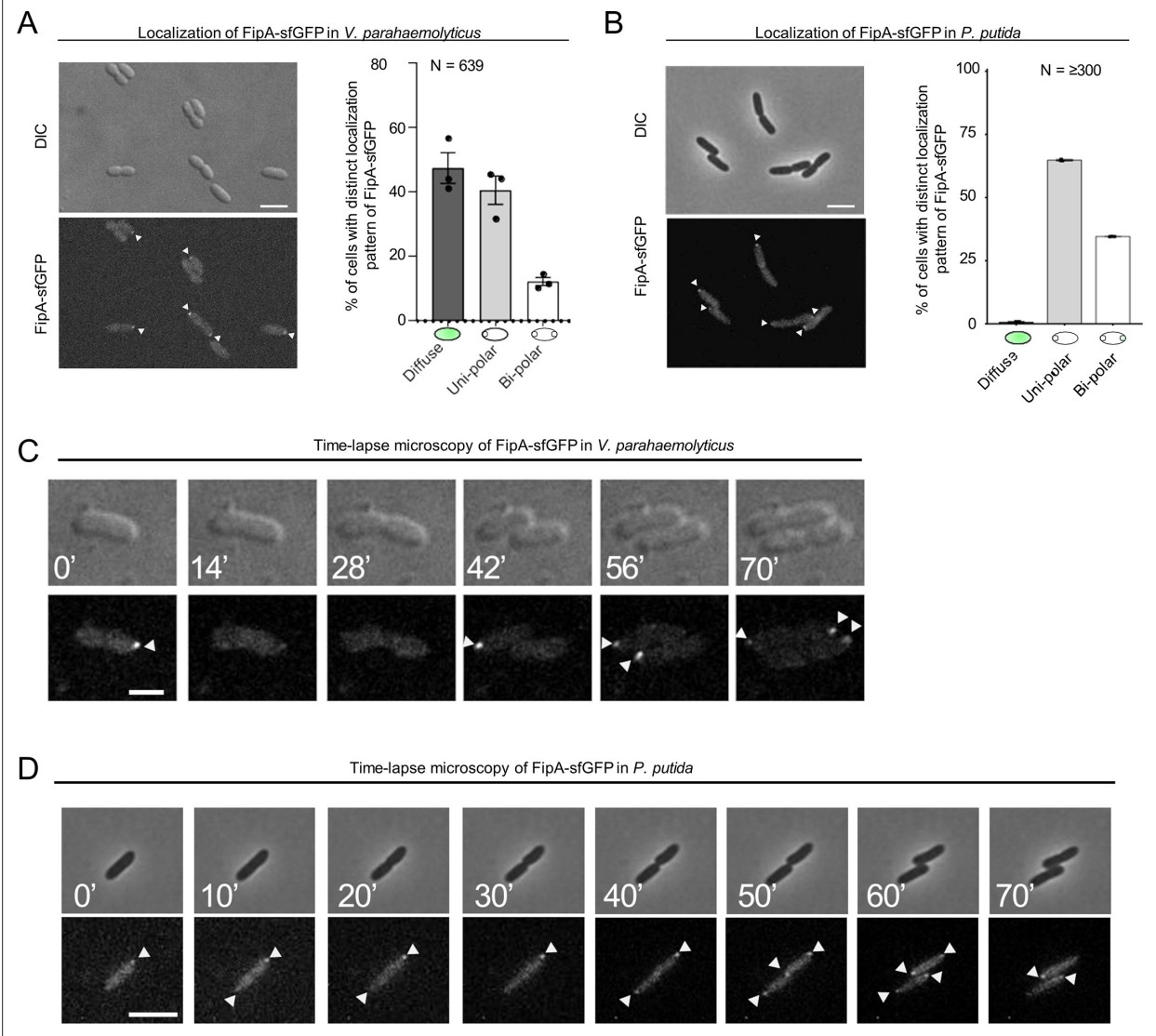

**Figure 6.** The localization pattern of FipA. (**A, B**) Localization pattern of fluorescently labeled FipA in *V. parahaemolyticus* and *P. putida*. (**A**) Representative micrographs of *V. parahaemolyticus* expressing FipA-sfGFP from its native promoter. Scale bar = 2 µm. The upper panel shows the DIC and the lower panel the corresponding fluorescence channel. To the right the localization was quantified accordingly. (**B**) The same analysis for *P. putida*. (**C, D**) Time lapse analysis of FipA-sfGFP localization over a cell cycle in *V. parahaemolyticus* (**C**) and *P. putida* (**D**). The numbers in the upper DIC micrographs show the minutes after start of the experiment. The scale bars equal 1 µm (**C**) and 5 µm (**D**).

The online version of this article includes the following source data and figure supplement(s) for figure 6:

**Figure supplement 1.** Production and function of FipA derivatives.

**Figure supplement 1—source data 1.** Scans of the original western blots for *Figure 6—figure supplement 1*.

**Figure supplement 1—source data 2.** Scans of the original western blots for *Figure 6—figure supplement 1* with labels.

**Figure supplement 2.** The localization pattern of FipA.

tagged with sfGFP (*Figure 6—figure supplement 1*). In wild-type strains, FipA localization remained diffuse in half of the population (*Figure 6A*; *Figure 6—figure supplement 1*). In the other half of the population, FipA formed distinct foci at the cell pole, either uni- or bi-polarly (*Figure 6A*; *Figure 6—figure supplement 2*). No delocalized foci were ever observed. The FipA localization behavior was explored further by following the protein through the cell cycle. We observed that FipA foci disappear frequently (*Figure 6C*; **14'**). A new focus appears at the opposite pole, sometimes before cell division (*Figure 6C*; **56'**), sometimes after (*Figure 6C*; **42'**). On rare cases, the first focus persists until after the second focus appears, resulting in bipolar foci (*Figure 6C*; **70'**). These results are consistent with

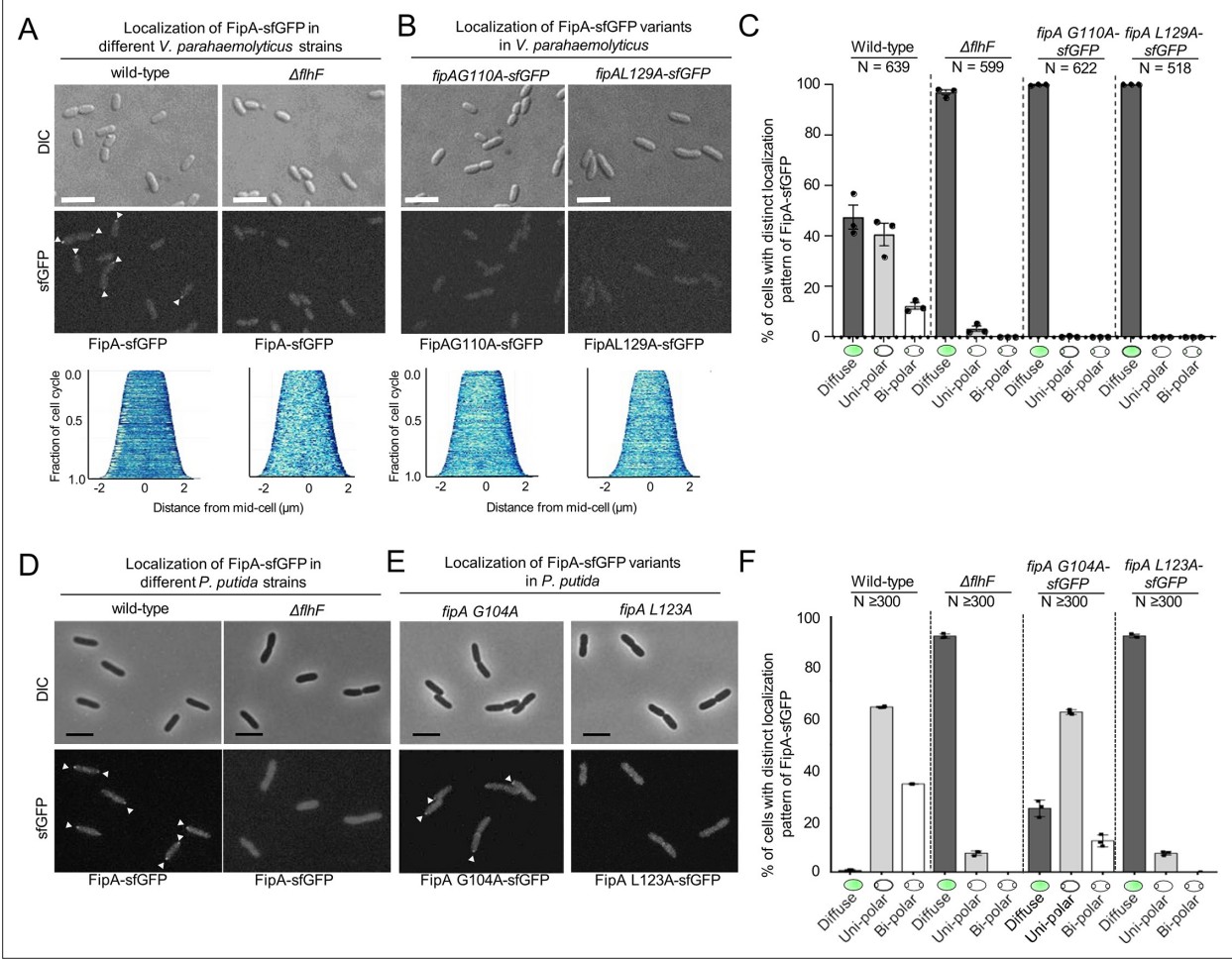

**Figure 7.** Normal localization of FipA depends on interaction with FlhF. (**A, B**) Localization pattern of *V. parahaemolyticus* FipA-sfGFP in the indicated wild-type and mutant strains. The upper panels display the DIC micrographs, the middle panel the corresponding fluorescence imaging (scale bar equals 5 μm), and the lower panel the corresponding demograph showing the fluorescence of FipA-sfGFP along the cell length. (**C**) Quantification of the cell localization pattern from the experiment shown in (**A, B**) as combined from three biological replicates. (**D, E, F**) The same analysis for the corresponding *P. putida* strains as indicated. The scale bar equals 5 μm. The data for *S. putrefaciens* is displayed in *Figure 7—figure supplement 3*.

The online version of this article includes the following figure supplement(s) for figure 7:

**Figure supplement 1.** Normal localization of FipA depends on interaction with FlhF.

**Figure supplement 2.** Normal localization of FipA depends on interaction with FlhF.

**Figure supplement 3.** Localization of *Sp*FipA in different strain backgrounds.

a protein that is recruited to the cell pole right before the start of the flagellum assembly after cell division, albeit the timing may vary between different species.

## Polar localization of FipA depends on the FipA-FlhF interaction

Finally, we determined whether FlhF plays a role in FipA localization. To this end, FlhF was deleted in the strains that are expressing FipA-sfGFP from its native promoter. Almost no FipA foci were detected in this background for both *V. parahaemolyticus* (*Figure 7A–C*; *Figure 7—figure supplement 1*) even though the protein levels of FipA were comparable to that in the wild type (*Figure 6—figure supplement 1*). Furthermore, the FipA variants that do not interact with FlhF (*Figure 5B*) were also labeled in the native *fipA* locus with sfGFP. Both *Vp*FipA G110A and *Vp*FipA L129A were incapable of forming polar foci (*Figure 7B and C*; *Figure 7—figure supplement 1*).

Altogether, these results suggest that, in *V. parahaemolyticus*, a direct interaction with FlhF, mediated by the residues in the DUF2802, is responsible for recruiting FipA to the pole.

## Functional conservation of FipA in *S. putrefaciens* and *P. putida*

Our FipA homology analyses (*Figure 1F*) indicated that the protein is widely conserved in species also possessing FlhF and FlhG. Accordingly, as in *V. parahaemolyticus*, FipA interacted with FlhF in the polarly flagellated gammaproteobacteria *S. putrefaciens*, which is monopolarly flagellated and *P. putida*, which is lophotrichously flagellated (*Figure 1—figure supplement 2*). We therefore asked to what extent the role of FipA is conserved for flagellation in these two species. To avoid interference with the secondary non-polar flagellar system of *S. putrefaciens*, we used a strain that is unable to form the secondary flagella (Δ*flaAB₂*). Determination of the flagellation state was carried out by fluorescence labeling of the flagellar filament(s) in both species.

### Spreading

As observed for *V. parahaemolyticus*, loss of FlhF and FipA negatively affected flagella-mediated spreading in soft agar and flagellation of *P. putida* and *S. putrefaciens*. However, for both species, the phenotype was not as severe as in *V. parahaemolyticus*: We observed about 40% and 50% in soft-agar spreading of Δ*flhF* mutants and about 25% and 90% spreading of Δ*fipA* mutants in *P. putida* and *S. putrefaciens*, respectively (*Figure 2A and C*; *Figure 2—figure supplement 1*). Accordingly, the mutations of *flhF* or *fipA* also negatively affected flagella number and localization in both species, with the exception of a *fipA* deletion in *S. putrefaciens*, where a dislocalization could not be observed (*Figure 2F and G*; *Figure 2—figure supplement 1*). Altogether, these results indicate that FipA is necessary for normal flagellation and swimming motility in all three model species, albeit at a different extent reaching from almost complete loss of swimming in *V. parahaemolyticus* to only a small effect in *S. putrefaciens*.

### Localization of FlhF

We then determined the localization of FlhF in mutants of FipA and HubP or FimV (the HubP homolog in *P. putida*) by using translational fusions of FlhF to fluorescent proteins expressed from the native site on the chromosome in *P. putida* and *S. putrefaciens* (*Figure 3—figure supplements 1 and 2*; *Rossmann et al., 2015*). As observed for *V. parahaemolyticus*, FipA and FimV were required for normal FlhF-mCherry localization in *P. putida* (*Figure 3E and F*; *Figure 3—figure supplement 3*), and FlhF was completely delocalized in the double mutant Δ*fipA* Δ*fimV*. In contrast, FlhF localization was not affected in a Δ*hubP* background in *S. putrefaciens* (*Figure 3—figure supplement 4*; *Rossmann et al., 2015*). However, in a *S. putrefaciens* mutant lacking *fipA*, monopolar localization of FlhF-mVenus decreased significantly (*Figure 3—figure supplement 4*). When both *hubP* and *fipA* were deleted in this species, only weak polar localization remained in a minority of cells. Frequently, polar fluorescence was completely lost and FlhF-mVenus was distributed in the cytoplasm. The differences in localization were not due to differences in abundance of FlhF (*Figure 3—figure supplement 1*).

The results showed that the function of FipA for flagellation and FlhF localization is generally conserved within the three different species. However, the polar landmark HubP appears to be involved in localizing FlhF, in particular in *S. putrefaciens*, where both HubP and FipA need to be deleted to delocalize FlhF from the cell pole.

### FipA localization pattern

As the next step, we determined the conservation of FipA domains' function and localization. We found that, as observed in *V. parahaemolyticus*, loss of the FipA transmembrane region (*fipA ΔTM*) phenocopied a Δ*fipA* mutant in *P. putida* and *S. putrefaciens* (*Figure 4C*, *Figure 2—figure supplement 1*). Furthermore, substitution of the conserved residues within the FipA DUF domain (G104A and L123A in *P. putida* FipA; G106A and L118A in *S. putrefaciens* FipA) decreased or abolished the interaction with FlhF but not FipA self-interaction (*Figure 5C and G*; *Figure 5—figure supplement 2*).

In *P. putida*, similar to *V. parahaemolyticus*, FipA formed distinct foci at the cell pole, either uni- or bi-polarly (*Figure 6A and B*). A remarkable difference between *P. putida* and *V. parahaemolyticus* is that in *P. putida*, FipA occurred as foci in virtually all cells (*Figure 6B*), whereas in *V. parahaemolyticus*, FipA remained diffuse in half of the population (*Figure 6A*). The ratio of bi-polar to uni-polar foci was also greater in *P. putida* (~1:2) than in *V. parahaemolyticus* (~1:5). Furthermore, in *P. putida*, the FipA foci were more stable, and foci at both poles often persisted until cell division. In fact, appearance

of the second focus at the new pole frequently occurred right after cell division (*Figure 6D*; **50'**), explaining the greater bi-polar to uni-polar ratio of FipA foci in this organism.

Finally, in *V. parahaemolyticus*, the localization but not the stability of FipA is dependent on the interaction with FlhF (*Figure 7*; *Figure 6—figure supplement 1*). This was similarly observed for *P. putida*, as the FipA L123A variant, which is unable to interact with FlhF (*Figure 1—figure supplement 2*), was also mostly distributed in the cytoplasm, whereas the variant FipA G104A did exhibit some polar localization, although reduced (*Figure 7E and F*; *Figure 7—figure supplement 2*). This is the variant that did show some interaction with *Pp*FlhF (*Figure 5C*). These results suggest that, in *V. parahaemolyticus* and *P. putida*, a direct interaction with FlhF, mediated by the residues in the DUF2802, is responsible for recruiting FipA to the pole.

Surprisingly, in *S. putrefaciens* mutants deleted in *flhF*, FipA displayed only a slight difference in polar localization (*Figure 7—figure supplement 3*). However, a *Sp*FipA G106A substitution reduced the protein's localization (to ~50% cells with foci). It is likely that in *S. putrefaciens*, HubP is (co-) mediating the localization of FipA, as in the absence of HubP, FipA-sfGFP localization at the cell pole drastically decreases (*Figure 7—figure supplement 3*). As observed for *V. parahaemolyticus* and *P. putida*, the stability of FipA was not dependent on the presence of FlhF or FipA in *S. putrefaciens* (*Figure 3—figure supplement 1*).

## Discussion

In bacteria, numerous processes are localized to specific cellular compartments. This is particularly evident in polarly flagellated bacteria, where the intricate flagellar multicomplex needs to be synthesized in a spatiotemporally regulated fashion. Flagellar synthesis is initiated by the assembly of the first flagellar building blocks within the cytoplasmic membrane, which in polar flagellates is targeted to the designated cell pole. In a wide range of bacterial species, the SRP-type GTPase FlhF and its antagonistic counterpart, the MinD-like ATPase FlhG, regulate flagellar positioning and number (*Kazmierczak and Hendrixson, 2013*; *Schuhmacher et al., 2015b*). In particular, FlhF has been shown to function as a positive regulator and a major localization factor for the initial flagellar building blocks of polar flagella. Upon production, GTP-bound dimeric FlhF localizes to the cell pole, but the mechanism by which the protein assumes its designated position at the cell pole remained elusive. One candidate protein, the polar landmark HubP (FimV in *Pseudomonas* sp.) had been identified earlier and was shown to interact with FlhF in *V. parahaemolyticus* (*Yamaichi et al., 2012*; *Dornes et al., 2024*). Accordingly, polar localization of FlhF is decreased in mutants lacking *hubP* (this study; *Rossmann et al., 2015*), however, in a number of cells FlhF localized normally and the cells showed normal flagellation. This was indicative that HubP does play a role in functionally recruiting FlhF, but that another factor was still missing. In this study, we identified the protein FipA as a second polarity factor for FlhF.

FipA consists of an N-terminal transmembrane domain and a cytoplasmic region with a conserved domain of unknown function (DUF2802). The corresponding gene is located immediately downstream of the motility operon that includes *flhF* and *flhG*. Notably, FipA is highly conserved among bacteria that use the FlhF/FlhG system to position the flagella, strongly suggesting that FipA has evolved together with FlhF and FlhG to regulate the flagellation pattern. In this study, we therefore investigated three distantly related γ-proteobacteria including one lophotrichous species, *V. parahaemolyticus*, *P. putida* and *S. putrefaciens*, in more detail. We found that FipA is conserved as a regulatory factor of FlhF and that its functions rely on similar features in all three species.

FipA is essential for normal flagellum synthesis, however, the severity of a *fipA* deletion differed among the three species studied. In *V. parahaemolyticus*, a *fipA* deletion phenocopies the loss of *flhF*, and cells no longer synthesize flagella. Notably, in the absence of FipA in *V. parahaemolyticus*, FlhF is still localized at the pole in about a third of the cells. Despite this, these cells were still unable to form flagella, strongly suggesting that occurrence of FlhF at the pole alone is not sufficient to trigger flagellum synthesis in the absence of FipA.

This strict requirement for FlhF has been reported in the closely related *V. alginolyticus* (*Kusumoto et al., 2009*). On the other hand, in *V. cholerae*, the requirement for FlhF on flagellation is less strict (*Green et al., 2009*), and in *P. putida* and *S. putrefaciens* even less so: mutants deleted in *flhF* or *fipA* of both species can generally still produce flagella and swim (this work; *Rossmann et al., 2015*). In *P. putida*, a Δ*fipA* mutant has a phenotype reminiscent of a Δ*flhF* mutant as both drastically decrease

the number of flagella, which may also be delocalized in *flhF* mutants (*Pandza et al., 2000*). In *S. putrefaciens*, deletion of *fipA* decreases the number of flagella to a similar extent as a *flhF* mutation, however, while in the latter case the flagella are frequently delocalized, the flagella remain at a polar postion, when *fipA* is missing. Taken together, these results strongly suggest that FipA does not act as a general polarity factor for the full flagellar machinery, but rather it stimulates the activity of FlhF to initiate polar flagellation.

## FipA and HubP affect polarity of FlhF through different mechanisms

In all three species, FipA directly interacts with FlhF as demonstrated by reciprocal co-immuno precipitations and by bacterial two-hybrid assays. These experiments suggest that FipA is able to dimerize (or oligomerize) and that the FlhF-FipA interaction is dependent on conserved residues in the DUF2802 domain. Furthermore, these residues are not required for the ability of FipA to interact with itself, suggesting that the two processes occur independently and at different protein interfaces. Microscopic and physiological assays showed that direct interaction between FlhF and FipA is required for FlhF targeting and function. In addition, a FipA mutant lacking its N-terminal transmembrane domain was non-functional, indicating that interaction of FlhF and FipA has to take place at the membrane.

In the absence of FipA, polar localization of FlhF was significantly decreased in all three species and almost completely diminished when *hubP* or *fimV* was deleted together with *fipA*. Removing HubP/FimV alone had a similar effect as removing FipA on the localization of FlhF, although to a lesser extent, as more cells still had polar FlhF in the Δ*hubP/fimV* mutant than in the Δ*fipA* mutant. Furthermore, the unipolar to bipolar ratio of FlhF foci increased in the Δ*hubP* mutant compared to the wildtype or the Δ*fipA* background. While the deletion of *fipA* increases the number of cells with diffuse localization, it does not affect the time frame in the cell cycle at which FlhF shifts from unipolar to bipolar in *V. parahaemolyticus*. In contrast, deleting *hubP* delays the time frame in which FlhF assumes a bipolar position. This indicates that HubP and FipA represent two different pathways that stimulate polar localization of FlhF, and that both are required to bring sufficient (active) FlhF molecules to trigger MS-ring formation and start flagellum assembly.

### Localization of FipA

A notable feature of FipA is its dynamic localization pattern within the cell over the cell cycle in all three species studied, the nature of which is unclear so far. Foci of FipA may appear mono- or bipolarly and frequently, but not necessarily, disappear once the flagellar apparatus, or the flagellar bundle in the case of *P. putida*, is established. It remains to be shown whether the FipA protein is actively moving from one pole to another or if newly formed FipA is recruited to the opposite pole while being degraded at the old position. Notably, in *V. parahaemolyticus* and *P. putida* FipA does not localize polarly in the absence of interaction with FlhF, suggesting that both proteins are recruited as a complex.

In contrast, in *S. putrefaciens,* FipA still localizes to the cell pole also in the absence of FlhF as long as the polar landmark HubP is present. This is indicating a more important role for HubP in this species. Accordingly, it has been shown recently that in *S. putrefaciens* HubP directly interacts with HubP via its NG domain to recruit the initial flagellar building blocks (*Dornes et al., 2024*). However, also in *S. putrefaciens* active FlhF localizes to the cell pole and flagella are formed in the absence of HubP. The nature of the polar marker that recruits or guides FipA or a FipA/FlhF to the designated pole is still elusive. The discovery of FipA provides a new point where spatiotemporal organization is coordinated. However, it remains to be explored if the complex is anchored through yet another protein, such as the TonB$_m$-PocA-PocB complex in *P. putida* (*Ainsaar et al., 2019*) or through an intrinsic feature of the cell envelope or cytoplasm during cell division.

### How does FipA upgrade the current model of polar flagellar synthesis?

Based on the findings in different species, our current model of FlhF/FlhG-mediated polar flagellar synthesis predicts, that upon production, GTP-bound dimeric FlhF is localizing or being recruited to the cell pole to where it recruits the first flagellar components to initiate the assembly. The monomeric form of the ATPase FlhG, the FlhF antagonist, is binding to the flagellar C-ring building block FliM and is transferred to the nascent flagellar structure, where it is released and dimerizes upon ATP binding (*Schuhmacher et al., 2015a*; *Blagotinsek et al., 2020*). The ATP-bound FlhG dimer can associate with

the membrane and interact with the GTP-bound FlhF dimer, thereby stimulating its GTPase activity. This leads to monomerization of FlhF and loss of polar localization (reviewed in *Kojima et al., 2020*; *Schuhmacher et al., 2015b*). Dimeric FlhG does also interact with the master regulator of flagella synthesis, FlrA (or FleQ in *Pseudomonas*) and prevents the synthesis of further early flagellar building blocks (*Dasgupta and Ramphal, 2001*; *Blagotinsek et al., 2020*; *Banerjee et al., 2021*). By this, FlhG links flagella synthesis with transcription regulation and effectively restricts to number of polar flagella that are formed. Based on this model, FipA may recruit FlhF to the membrane and stimulate or stabilize GTP binding and dimerization of FlhF to promote polar localization and initiation of flagella synthesis. Thus, the role of FipA is to shift the equilibrium to the active state of FlhF in order to start assembly of the MS ring. Accordingly, current studies address the structural basis of the FipA-FlhF interaction, the potential effect of FipA on GTP-binding and dimerization of FlhF, and the role of the polar marker HubP in this process.

In addition, *fipA* expression is independent of the rest of the flagellar genes in other species of *Vibrio* (*Moisi et al., 2009*; *Petersen et al., 2021*) and *S. putrefaciens* (*Schwan et al., 2022*). Thus, *fipA* transcription could provide another regulatory point to lead to the synthesis of a flagellum. Thus, the discovery of FipA opens several novel open questions in the field of flagellum regulation.

# Materials and methods
## Growth conditions and media
In all experiments, *V. parahaemolyticus* and *E. coli* were grown in LB medium or on LB agar plates at 37 °C containing antibiotics in the following concentrations: 50 µg/mL kanamycin, 100 µg/mL ampicillin and 20 µg/mL chloramphenicol for *E. coli* and 5 µg/mL for *V. parahaemolyticus*. *S. putrefaciens* CN-32 and *P. putida* KT 2440 were grown in LB medium or LB agar plates at 30 °C. When required, the media were supplemented with 50 µg/mL kanamycin, 300 µM 2,6-diaminopimelic acid and/or 10% (w/v) sucrose.

## Strains and plasmids
The strains and plasmids used in this study are listed in *Supplementary file 1c and d*, respectively. Primers used are listed in *Supplementary file 1e*. *E. coli* strain SM10 $\lambda$ *pir* was used to transfer DNA into *V. parahaemolyticus* by conjugation (*Miller and Mekalanos, 1988*). For DNA transfer into *S. putrefaciens* and *P. putida*, *E. coli* WM3064 was used. *E. coli* strains DH5α $\lambda$ *pir* and SM10 $\lambda$ *pir* were used for cloning. Construction of *V. parahaemolyticus* deletion mutants was performed with standard allele exchange techniques using derivatives of plasmid pDM4 (*Donnenberg and Kaper, 1991*). Chromosomal deletions and integrations in *S. putrefaciens* and *P. putida* were carried out by sequential crossover as previously described (*Rossmann et al., 2015*) using derivatives of plasmid pNPTS138-R6K (*Lassak et al., 2010*).

## Construction of plasmids
*Plasmid pSW022*: The regions flanking *vp2224* (*fipA*) were cloned with primers VP2224-del-a/ b and VP2224-del-c/ d, using *V. parahaemolyticus* RIMD 2210633 chromosomal DNA as template. The resulting products were fused in a third PCR using primers VP2224-del-a/ VP2224-del-d. The end product was digested with XbaI and ligated in the equivalent site of vector pDM4, resulting in plasmid pSW022. The mutation in *V. parahaemolyticus* was confirmed with a PCR using primers VP2224-del-a/VP2224-check. *Plasmid pPM123:* The regions flanking aa 7–27 of *vp2224* (*fipA*) were cloned with primers VP2224-del-a/ del AA7-27 vp2224-b and del AA7-27 vp2224-c/ VP2224-del-d, using *V. parahaemolyticus* RIMD 2210633 chromosomal DNA as template. The resulting products were fused in a third PCR using primers VP2224-del-a/ VP2224-del-d. The end product was digested with XbaI and ligated in the equivalent site of vector pDM4, resulting in plasmid pPM123. The mutation in *V. parahaemolyticus* was confirmed with a PCR using primers VP2224-del-a/VP2224-check. *Plasmid pPM178:* The gene *vp2224* (*fipA*) was amplified from *V. parahaemolyticus* RIMD 2210633 chromosomal DNA with primers C-term sfGFP-vp2224-a/-b; the downstream region with primers C-term sfGFP-vp2224-e/f; the gene encoding sfGFP with C-term sfGFP-vp2224-c/d from plasmid pJH036. The three products were fused together in another PCR using primers C-term sfGFP-vp2224-a/f. The obtained product, encoding FipA fused in frame to sfGFP via a 5-residue

linker, was digested with SpeI and SphI and ligated in vector pDM4 digested with XbaI and SphI, resulting in plasmid pPM178. The mutation in *V. parahaemolyticus* was confirmed with a PCR using primers C-term sfGFP-vp2224-f/VP2224-check. *Plasmid pPM179 & pPM180:* The region upstream of *vp2224* (*fipA*) were amplified as for pSW022. The downstream region was amplified with downstream vp2224-cw/VP2224-del-d from *V. parahaemolyticus* RIMD 2210633. The gene *vp2224* itself was amplified with primers vp2224 cw restore deletion/vp2224 ccw restore deletion, from plasmids pEP005 & pEP006, carrying the mutations G110A and L129A. The products were fused in a third PCR using primers VP2224-del-a/ VP2224-del-d. The end product was digested with XbaI and ligated in the equivalent site of vector pDM4, resulting in plasmids pPM179 & pPM180. The the re-insertion in *V. parahaemolyticus* SW01 was confirmed with a PCR using primers VP2224-del-a/VP2224-check. *Plasmid pPM187 & pPM191:* The insertion of sfGFP at the C-terminus of FipA was cloned with the same strategy for pPM178, but using pPM179 & pPM180 as templates for the *vp2224* sequence, resulting in plasmids pPM191 & pPM187, respectively. *Plasmid pPM146:* The gene *vp2224* (*fipA*) was amplified from *V. parahaemolyticus* RIMD 2210633 chromosomal DNA with primers vp2224-cw-pBAD/vp2224-ccw-pBAD. The product was digested with enzymes XbaI and SphI, and inserted in the corresponding site in plasmid pBAD33. *Plasmid pPM159:* The gene *vp2224* (*fipA*) was amplified from *V. parahaemolyticus* RIMD 2210633 chromosomal DNA with primers C-term sfGFP-vp2224-a/-b, and the gene encoding sfGFP with C-term sfGFP-vp2224-c/sfGFP-1-ccw from plasmid pJH036. The resulting products were fused in another reaction with primers C-term sfGFP-vp2224-a/sfGFP-1-ccw, and this final product was digested with XbaI and cloned in pBAD33. *Plasmid pPM194:* Same as pPM159, but *vp2224 ΔTM* was amplified from pPM123 using primers vp2224 AA1-6/28-end w/o Stop/sfGFP-1-ccw. *Plasmids pSW74 & pSW119:* The region of the gene *vp2224* (*fipA*) encoding the cytoplasmic part (residues 28–163) was amplified with primers tr2224 put18C cw/ pUT18C/pKT25-vp2224-ccw from *V. parahaemolyticus* chromosomal DNA. The product was digested with KpnI and XbaI and ligated in the corresponding site in pKT25, to generate pSW74, and in pUT18C, to generate pSW119. *Plasmid pPM118 & pPM119:* The cytoplasmic part of the gene *vp2224* (*fipA*) was amplified with primers pUT18/pKNT25-tr-vp2224-cw & pUT18/pKNT25-vp2224ccw from *V. parahaemolyticus* chromosomal DNA. The product was digested with KpnI and XbaI and ligated in the corresponding site in pUT18, to generate pPM118, and in pKNT25, to generate pPM119. *Plasmid pPM124 & pPM128:* The gene *vp2234* (*flhF*) was amplified with primers pUT18/pKNT25- vp2234-cw & pUT18/pKNT25-vp2234 -ccw from *V. parahaemolyticus* chromosomal DNA. The product was digested with KpnI and XbaI and ligated in the corresponding site in pUT18, to generate pPM124, and in pKNT25, to generate pPM128. *Plasmid pPM132 & pPM136:* The gene *vp2234* (*flhF*) was amplified with primers pUT18C/pKT25- vp2234-cw & pUT18C/pKT25-vp2234 -ccw from *V. parahaemolyticus* chromosomal DNA. The product was digested with KpnI and XbaI and ligated in the corresponding site in pUT18C, to generate pPM132, and in pKT25, to generate pPM136. *Plasmids pPM160, pPM161 & pPM162:* Site directed mutagenesis was performed on plasmid pSW74 by the QuickChange method (*Zheng et al., 2004*), using primers vp2224-Gly-110Ala-cw/ccw, vp2224-Glu 126Ala-cw/ccw or vp2224-Leu 129Ala-cw/ccw. After digesting the template with DpnI, the products were transformed into *E. coli* and the mutations were confirmed by sequencing. *Plasmid pPM106:* The gene vp2224 (fipA) was amplified with primers pUT18C/pKT25-vp2224-cw & pUT18C/pKT25-vp2224ccw from V. parahaemolyticus chromosomal DNA. The product was digested with KpnI and XbaI and ligated in the corresponding site in pKTop, to generate pPM106. *Plasmid pPM109 & pPM112:* The *phoA-lacZα* fragment of pKTop was amplified with primers vp2224 C-term PhoA-LacZ cw & end -LacZ w/o STOP ccw. The full length *vp2224(fipA)* gene was amplified from plasmid pPM146 using primers LacZ to vp2224 w/o ATG & end vp2224 ccw. The *vp2224 ΔTM* allele was amplified from plasmid pPM123 using primers vp2224 AA1-6/28-end w/o Stop & end vp2224 ccw. The PCR products were fused in a second PCR using primers vp2224 C-term PhoA-LacZ cw & end vp2224 ccw. The fusion product was digested with XbaI and HindIII and cloned in the corresponding site of plasmid pBAD33.

Plasmids for genetic manipulation of *S. putrefaciens* and *P. putida:* The desired DNA fragments were generated by PCR using appropriate primer pairs (see *Supplementary file 1e*) that in addition create overhangs suitable for subsequent Gibson assembly into EcoV-digested vector pNTPS138-R6K (*Gibson et al., 2009*; *Lassak et al., 2010*). If necessary, primer overhangs were also used to generate base substitutions in the gene fragment to be cloned. *Plasmids for Bacterial Two Hybrid (BACTH)*

*analysis* were similarly generated by amplification of the desired DNA-fragments, which were then introduced into the suitable vectors by Gibson assembly.

## Soft-agar swimming assays

*V. parahaemolyticus* soft-agar swimming assays were essentially performed as described in *Ringgaard et al., 2014* with the following modifications. Late exponential cultures of the required strains were used to prick LB plates with 0.3% agar, and incubated for 30 hr at 30 °C. For *S. putrefaciens* and *P. putida* soft-agar swimming assays, 2 µl of an exponentially growing culture of the appropriate strains were spotted onto 0.25% LB agar plates and incubated at 30 °C (*S. putrefaciens*) or room temperature (*P. putida*) for about 18 hrs. Strains to be directly compared were always spotted onto the same plate. For the swimming assays in soft agar, strains of *V. parahaemolyticus* and *S. putrefaciens*, which are lacking lateral flagellin gene(s) (Δ*lafA*; Δ*flaAB₂*) were used.

## Bacterial-two-hybrid experiments, BACTH

BTH101 cells were made competent with calcium chloride. 15 µL aliquots were spotted in a 96 well plate, to which 2.5 µL of the corresponding pUT18(C) and pK(N)T25 derivative plasmids were added. After 30 min on ice, a heat shock at 42 °C was applied for 30 s. The transformed cells were allowed to recover for 1 hr, after which the were grown in selective LB broth for another 3 hr. The resulting cultures were spotted on plates containing kanamycin, ampicillin, IPTG (0.25 mM) and X-Gal (80 µg/mL). The plates were photographed after 48–72 hr at 30 °C.

## Video tracking of swimming cells

Video tracking of swimming cells was performed essentially as described previously (*Ringgaard et al., 2014*). Swimming cells were recorded using the streaming acquisition function in the Metamorph software and the swimming paths of individual cells were tracked using the MTrackJ plug-in for ImageJ. The swimming speed, displacement, and number of reversals of individual cells were then measured and the average plotted with error bars indicating the standard deviation. Video tracking was performed using a Zeiss Axio Imager M1 fluorescence microscope. Images were collected with a Cascade:1 K CCD camera (Photometrics), using a Zeiss αPlan-Fluar 40 x/1.45 Oil phase contrast objective.

## Transmission-electron-microscopy (TEM) analysis

Cell cultures grown to an OD600=0.5–0.6 were spotted on a plasma-discharged carbon-coated copper grid (Plano, Cat#S162-3) and rinsed with 0.002% uranyl acetate. Afterwards they were rinsed with water and blotted dry with Whatman filter paper. TEM images were obtained with a JEOL JEM-1400 Plus 120 KV transmission electron microscope at 80 kV.

## Flagellum labeling

To fluorescently label flagellar filaments, maleimide-ligates dyes (Alexa Fluor 488 C5 maleimide fluorescent dye; Thermo Fisher Scientific) were coupled to surface-exposed cysteine residues, which were introduced into the flagellins (FlaA and FlaB) of the polar flagellar system in *S. putrefaciens* and *P. putida* as described (*Kühn et al., 2017*; *Hintsche et al., 2017*). Images were recorded as described in the microscopy section.

## Fluorescence microscopy

Florescence microscopy on *V. parahaemolyticus* was carried out essentially as previously described (*Muraleedharan et al., 2018*), using a Nikon eclipse Ti inverted Andor spinning-disc confocal microscope equipped with a 100 x lens, an Andor Zyla sCMOS cooled camera, and an Andor FRAPPA system. Fluorescence microscopy on *S. putrefaciens* and *P. putida* was carried out as previously described (*Kühn et al., 2017*) using a custom microscope set-up (Visitron Systems, Puchheim, Germany) based on a Leica DMI 6000 B inverse microscope (Leica) equipped with a pco.edge sCMOS camera (PCO), a SPECTRA light engine (lumencor), and an HCPL APO 63×/1.4–0.6 objective (Leica) using a custom filter set (T495lpxr, ET525/50 m; Chroma Technology) using the VisiView software (Visitron Systems, Puchheim, Germany). Microscopy images were analyzed using ImageJ imaging software (http://rsbweb.nih.gov/ij) and Metamorph Offline (version 7.7.5.0, Molecular Devices). DIC (*Vibrio*)

and phase contrast (*Shewanella*, *Pseudomonas*) were used according to the preferred settings for the corresponding species in the labs. Demographs were generated as described by *Cameron et al., 2014* modified in *Heering and Ringgaard, 2016* and *Heering et al., 2017*.

## Mapping interaction partners using co-immunoaffinity purification and mass spectrometry (co-IP-MS)

For sample preparation of the IP-MS experiments we were using a modified version of the protocol presented by *Turriziani et al., 2014*. Cells were centrifuged and cell pellets were washed with cold PBS. Cell pellets were resuspended in lysis buffer (50 mM HEPES (pH 7.5), 150 mM NaCl, 0.5 % NP50, 5 mM EDTA, compete mini protease inhibitors (Complete Mini (Roche)). Cell lysis was performed by repetitive sonication. After removing cell debris by centrifugation 10 µl GFP-trap Sepharose (Chromotek) slurry was added to the lysate and incubation was carried out for 1.5 hr on a rotating shaker at 4 °C. Then the beads were pelleted, the supernatant removed and the beads washed 4 x with 100 mM $NH_4CO_3$ to remove remaining protease inhibitors and detergents. 200 µl elution buffer (1 M urea, 100 mM $NH_4CO_3$, 1 µg Trypsin (Promega)) was added to the beads and incubated at 1000 rpm on a thermomixer at 27 °C for 45 min. Beads were centrifuged and supernatant was collected. In order to increase peptide recovery 2 washing steps with 100 µL elution buffer 2 (1 M urea, 100 mM $NH_4CO_3$, 5 mM Tris(2-caboxyethyl)phosphine)) was performed and the individual bead supernatants were collected into one tube. Tryptic digestion was carried out overnight at 30 °C. After digest alkylation was performed with 10 mM iodoacetamide at 25 °C (in the dark). Then the samples were acidified (1% trifluoroacetic acid [TFA]) and C18 Microspin column (Harvard Apparatus) was carried out according to the manufacturer´s instruction. The samples were dried and recovered in 0.1% TFA and applied to liquid chromatography-mass spectrometry (LC-MS) analysis.

LC-MS analysis of digested lysates was performed on a Thermo QExactive Plus mass spectrometer (Thermo Scientific), which was connected to an electrospray ionsource (Thermo Scientific). Peptide separation was carried out using an Ultimate 3000 RSLCnano with Proflow upgrade (Thermo Scientific) equipped with a RP-HPLC column (75 µm x 42 cm) packed in-house with C18 resin (2.4 µm; Dr. Maisch) on an in-house designed column heater. The following separating gradient was used: 98% solvent A (0.15% formic acid) and 2% solvent B (99.85% acetonitrile, 0.15% formic acid) to 35% solvent B over 90 at a flow rate of 300 nl/min. The data acquisition mode was set to obtain one high resolution MS scan at a resolution of 70,000 full width at half maximum (at m/z 200) followed by MS/MS scans of the 10 most intense ions. To increase the efficiency of MS/MS attempts, the charged state screening modus was enabled to exclude unassigned and singly charged ions. The dynamic exclusion duration was set to 30 s. The ion accumulation time was set to 50ms (MS) and 50ms at 17,500 resolution (MS/MS). The automatic gain control (AGC) was set to $3x10^6$ for MS survey scan and $1x10^5$ for MS/MS scans.

MS raw data was then analyzed with MaxQuant (Version 1.6.3.4; https://www.nature.com/articles/nbt.1511) using a *V.parahaemolyticus* RIMD 2210633 uniprot database (https://www.uniprot.org/). MaxQuant was executed in standard settings with activated 'match between runs' option. The search criteria were set as follows: full tryptic specificity was required (cleavage after lysine or arginine residues); two missed cleavages were allowed; carbamidomethylation (C) was set as fixed modification; oxidation (M) and deamidation (N,Q) as variable modification. For further data analysis the MaxQuant LFQ values were loaded into Perseus (https://www.nature.com/articles/nmeth.3901) and a Student's T-test was performed on LFQ values with false discovery rate 0.01 and S0: 0.1 as significance cut-off. The mass spectrometry proteomics data have been deposited to the ProteomeXchange Consortium via the PRIDE (PubMed ID: 34723319) partner repository with the dataset identifier PXD045379 (http://www.ebi.ac.uk/pride).

## Membrane topology mapping of FipA

We experimentally determined the membrane orientation of FipA in the membrane using the dual pho-lac reporter system (*Karimova et al., 2009*), which consists of a translational fusion of the *E. coli* alkaline phosphatase fragment PhoA22-472 and the α-peptide of *E. coli* β-galactosidase, LacZ4-60. A periplasmic localization of the reporter leads to high alkaline phosphatase activity and low β-galactosidase activity, whereas a cytosolic location of the reporter results in high β-galactosidase activity and low alkaline phosphatase activity. Pho-Lac-FipA and FipA-Pho-Lac fusion proteins were ectopically

expressed in *E. coli* DH5α grown on a dual-indicator LB medium containing a blue indicator for phosphatase activity (X-Phos) and red indicator for β-galactosidase activity (Red-Gal) (see *Figure 1—figure supplement 1*).

## Bioinformatic analysis

Homologues of FipA were searched using BLAST against the KEGG database, with the sequence of the *V. parahaemolyticus* protein. All homologues containing a DUF2802 were included. The search was later expanded to species known to encode a FlhF homologue (defined as the highest scoring result from a BLAST search using FlhF of *V. parahaemolyticus* as a query, that was also encoded upstream of an FlhG homologue). The flagellation phenotype was later corroborated in the description registered at the List of Prokaryotic names with Standing in Nomenclature (*Parte et al., 2020*).

## Acknowledgements

We are grateful to Dr. Kathrin Schirner for comments on the manuscript and suggestions for experiments, and Manuel González-Vera for his help on the phylogenetic search. We would like to thank Ulrike Ruppert for great technical support and Jan Heering for construction of plasmid pJH036. This work was supported by the Ludwig-Maximilians-Universität München and the Max Planck Society (SR) and by a grant (TRR 174 P12) from the Deutsche Forschungsgemeinschaft DFG to KMT within the framework of the DFG priority program TRR 174.

## Additional information

### Funding

| Funder | Grant reference number | Author |
| --- | --- | --- |
| Max-Planck-Gesellschaft | | Erick E Arroyo-Pérez<br>Alejandra Alvarado<br>Stephan Wimmi<br>Timo Glatter<br>Simon Ringgaard |
| Deutsche Forschungsgemeinschaft | TRR 174-P12 | John C Hook<br>Kai Thormann |
| Ludwig-Maximilians-Universität München | | Simon Ringgaard |

The funders had no role in study design, data collection and interpretation, or the decision to submit the work for publication.

### Author contributions

Erick E Arroyo-Pérez, Conceptualization, Data curation, Formal analysis, Investigation, Visualization, Methodology, Writing – original draft, Writing – review and editing; John C Hook, Conceptualization, Formal analysis, Validation, Investigation, Visualization, Methodology, Writing – original draft, Writing – review and editing; Alejandra Alvarado, Data curation, Validation, Investigation, Methodology, Writing – review and editing; Stephan Wimmi, Investigation, Methodology, Writing – review and editing; Timo Glatter, Data curation, Formal analysis, Validation, Investigation, Methodology; Kai Thormann, Conceptualization, Supervision, Funding acquisition, Investigation, Writing – original draft, Project administration, Writing – review and editing; Simon Ringgaard, Conceptualization, Funding acquisition, Validation, Investigation, Visualization, Writing – original draft, Project administration, Writing – review and editing

### Author ORCIDs

Kai Thormann ⓘ https://orcid.org/0000-0001-7292-4884

Reviewer #1 (Public review): https://doi.org/10.7554/eLife.93004.3.sa1
Reviewer #2 (Public review): https://doi.org/10.7554/eLife.93004.3.sa2

Reviewer #3 (Public review): https://doi.org/10.7554/eLife.93004.3.sa3
Author response https://doi.org/10.7554/eLife.93004.3.sa4

## Additional files

### Supplementary files

• Supplementary file 1. Supplementary tables. (**a**) Enriched proteins in Co-IP FlhF-sfGFP vs. sfGFP. The genes with the largest enrichment in FlhF-sfGFP vs. sfGFP are shown. The corresponding protein designation is indicated if available. (**b**) Flagellation pattern and presence of FipA and FlhF in bacteria. (**c**) Bacterial strains used in this study. (**d**) Plasmids used in this study. (**e**) Oligonucleotides used in this study.

• MDAR checklist

• Source data 1. Contains the data used for generation of the indicated figure panels.

### Data availability

All data generated or analysed during this study are included in the manuscript and supporting files.

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

# Appendix 1

## Appendix 1—key resources table

| Reagent type (species) or resource | Designation | Source or reference | Identifiers | Additional information |
|---|---|---|---|---|
| Gene (Vibrio parahaemolyticus) | *fipA* | KEGG | VP2224 | |
| Gene (Pseudomonas putida) | *fipA* | KEGG | PP_4331 | |
| Gene (Shewanella putrefaciens) | *fipA* | KEGG | Sputcn32_2550 | |
| Gene (Vibrio parahaemolyticus) | *flhF* | KEGG | VP2234 | |
| Gene (Pseudomonas putida) | *flhF* | KEGG | PP_4343 | |
| Gene (Shewanella putrefaciens) | *flhF* | KEGG | Sputcn32_2561 | |
| Strain (Escherichia coli) | SM10 λ pir | *Simon et al., 1983* | | |
| Strain (Escherichia coli) | DH5pir | *Miller and Mekalanos, 1988* | | |
| Strain (Escherichia coli) | BTH101 | Euromedex | | |
| Strain (Escherichia coli) | WM3064 | Metcalf, University of Illinois, Urbana-Champaign | | |
| Strain (Vibrio parahaemolyticus) | RIMD 2210633 | *Makino et al., 2003* | | Wild type |
| Strain (Vibrio parahaemolyticus) | Δvp2234 (ΔflhF), Δvpa1548 (ΔlafA) | *Arroyo-Pérez and Ringgaard, 2021* | EP12 | |
| Strain (Vibrio parahaemolyticus) | Δvp2225 (ΔcheW) | *Ringgaard et al., 2014* | SR58 | |
| Strain (Vibrio parahaemolyticus) | Δvpa1548 (ΔlafA) | *Arroyo-Pérez and Ringgaard, 2021* | JH2 | |
| Strain (Vibrio parahaemolyticus) | Δvp2224 (ΔfipA), Δvpa1548 (ΔlafA) | this study | EP15 | Materials and Methods |
| Strain (Vibrio parahaemolyticus) | Δvp2234 (ΔflhF) | *Arroyo-Pérez and Ringgaard, 2021* | PM60 | |
| Strain (Vibrio parahaemolyticus) | Δvp2224 (ΔfipA) | this study | SW01 | Materials and Methods |
| Strain (Vibrio parahaemolyticus) | Δvp2191 (ΔhubP) | *Arroyo-Pérez and Ringgaard, 2021* | JH4 | |
| Strain (Vibrio parahaemolyticus) | Δvp2234::vp2234-sfgfp (ΔflhF::flhF-sfgfp) | *Arroyo-Pérez and Ringgaard, 2021* | PM69 | |
| Strain (Vibrio parahaemolyticus) | Δvp2234::vp2234-sfgfp (ΔflhF::flhF-sfgfp), Δvp2224 (fipA) | this study | PM77 | Materials and Methods |
| Strain (Vibrio parahaemolyticus) | Δvp2234::vp2234-sfgfp (ΔflhF::flhF-sfgfp), Δvp2191 (hubP) | *Arroyo-Pérez and Ringgaard, 2021* | EP11 | |
| Strain (Vibrio parahaemolyticus) | Δvp2234::vp2234-sfgfp (ΔflhF::flhF-sfgfp), Δvp2224 (fipA), Δvp2191 (hubP) | this study | EP09 | Materials and Methods |
| Strain (Vibrio parahaemolyticus) | vp2224 (fipA) L129A | this study | PM65 | Materials and Methods |
| Strain (Vibrio parahaemolyticus) | vp2224 (fipA) G110A | this study | PM66 | Materials and Methods |
| Strain (Vibrio parahaemolyticus) | Δvp2234::vp2234-sfgfp (ΔflhF::flhF-sfgfp), vp2224 (fipA) L129A | this study | EP16 | Materials and Methods |
| Strain (Vibrio parahaemolyticus) | Δvp2234::vp2234-sfgfp (ΔflhF::flhF-sfgfp), vp2224 (fipA) G110A | this study | EP17 | Materials and Methods |
| Strain (Vibrio parahaemolyticus) | vp2224 L129A, Δvpa1548(lafA) | this study | EP13 | Materials and Methods |
| Strain (Vibrio parahaemolyticus) | vp2224 G110A, Δvpa1548(lafA) | this study | EP14 | Materials and Methods |
| Strain (Vibrio parahaemolyticus) | Δvp2224::vp2224-sfgfp (ΔfipA::fipA-sfgfp) | this study | PM64 | Materials and Methods |
| Strain (Vibrio parahaemolyticus) | Δvp2224::vp2224-sfgfp (ΔfipA::fipA-sfgfp), Δvp2234 (flhF) | this study | PM68 | Materials and Methods |
| Strain (Vibrio parahaemolyticus) | Δvp2224::vp2224 L129A-sfgfp (ΔfipA L129A::fipA-sfgfp) | this study | PM71 | Materials and Methods |
| Strain (Vibrio parahaemolyticus) | Δvp2224::vp2224 G110A-sfgfp (ΔfipA G110A::fipA-sfgfp) | this study | PM72 | Materials and Methods |
| Strain (Pseudomonas putida) | KT2440 | *Nelson et al., 2002* | | Wild type |

*Appendix 1 Continued on next page*

*Appendix 1 Continued*

| Reagent type (species) or resource | Designation | Source or reference | Identifiers | Additional information |
|---|---|---|---|---|
| Strain (Pseudomonas putida) | FliC S267C | *Hintsche et al., 2017* | | |
| Strain (Pseudomonas putida) | FliC S267C ΔflhF | this study | | Materials and Methods |
| Strain (Pseudomonas putida) | ΔfipA | this study | | Materials and Methods |
| Strain (Pseudomonas putida) | FliC S267C ΔfipA | this study | | Materials and Methods |
| Strain (Pseudomonas putida) | FipA -DILEL-sfGFP | this study | | Materials and Methods |
| Strain (Pseudomonas putida) | FliC S267C ΔflhF FipA-DILEL-sfGFP | this study | | Materials and Methods |
| Strain (Pseudomonas putida) | FlhF-GS-mCherry | this study | | Materials and Methods |
| Strain (Pseudomonas putida) | FliC S267C ΔfimV FipA -DILEL-sfGFP | this study | | Materials and Methods |
| Strain (Pseudomonas putida) | ΔfipA FlhF-GS-mCherry | this study | | Materials and Methods |
| Strain (Pseudomonas putida) | FliC S267C FipA G104A | this study | | Materials and Methods |
| Strain (Pseudomonas putida) | FipA L123A | this study | | Materials and Methods |
| Strain (Pseudomonas putida) | FliC S267C FipA L123A | this study | | Materials and Methods |
| Strain (Pseudomonas putida) | FliC S267C FipA-DILEL-sfGFP | this study | | Materials and Methods |
| Strain (Pseudomonas putida) | FliC S267C FipA L116A | this study | | Materials and Methods |
| Strain (Pseudomonas putida) | FliC S267C FipA G104A-DILEL-sfGFP | this study | | Materials and Methods |
| Strain (Pseudomonas putida) | FliC S267C FipA L123A-DILEL-sfGFP | this study | | Materials and Methods |
| Strain (Pseudomonas putida) | FliC S267C FipA L116A-DILEL-sfGFP | this study | | Materials and Methods |
| Strain (Pseudomonas putida) | FlhF-GS-mCherry ΔfimV | this study | | Materials and Methods |
| Strain (Pseudomonas putida) | FliC S267C ΔfipA FipA KI | this study | | Materials and Methods |
| Strain (Pseudomonas putida) | FlhF-GS-mCherry ΔfimV ΔfipA | this study | | Materials and Methods |
| Strain (Pseudomonas putida) | FlhF-GS-mCherry FipA L123A | this study | | Materials and Methods |
| Strain (Pseudomonas putida) | FlhF-GS-mCherry FipA L116A | this study | | Materials and Methods |
| Strain (Pseudomonas putida) | FlhF-GS-mCherry FipA G104A | this study | | Materials and Methods |
| Strain (Pseudomonas putida) | FliC S267C FlhF D362A-GS-mCherry | this study | | Materials and Methods |
| Strain (Pseudomonas putida) | FliC S267C FlhF-GS-mCherry FipA ΔTMD | this study | | Materials and Methods |
| Strain (Pseudomonas putida) | FliC S267C FlhF-GS-mCherry FipA ΔTMD | this study | | Materials and Methods |
| Strain (Pseudomonas putida) | FliC S267C FipA ΔTMD-DILEL-sfGFP | this study | | Materials and Methods |
| Strain (Shewanella putrefaciens) | CN-32 | *Fredrickson et al., 1998* | | Wild type |
| Strain (Shewanella putrefaciens) | ΔflhF | *Rossmann et al., 2015* | | |
| Strain (Shewanella putrefaciens) | CheA-mCherry | this study | | Materials and Methods |
| Strain (Shewanella putrefaciens) | flgE1 T183C | *Rossmann et al., 2019* | | |
| Strain (Shewanella putrefaciens) | flaB1 T166C flaA1 T174C ΔflagL | *Kühn et al., 2017* | | |
| Strain (Shewanella putrefaciens) | ΔfipA | this study | | Materials and Methods |
| Strain (Shewanella putrefaciens) | CheA-mCherry ΔfipA | this study | | Materials and Methods |
| Strain (Shewanella putrefaciens) | flaB1 T166C flaA1 T174C ΔflagL ΔfipA | this study | | Materials and Methods |
| Strain (Shewanella putrefaciens) | FlhF-GS-mVenus | this study | | Materials and Methods |
| Strain (Shewanella putrefaciens) | FipA-DILEL-sfGFP | this study | | Materials and Methods |
| Strain (Shewanella putrefaciens) | FlhF-GS-mVenus ΔfipA | this study | | Materials and Methods |
| Strain (Shewanella putrefaciens) | FlhF-GS-mVenus ΔflhG | this study | | Materials and Methods |
| Strain (Shewanella putrefaciens) | FipA-DILEL-sfGFP ΔflhF | this study | | Materials and Methods |
| Strain (Shewanella putrefaciens) | FlhF-GS-mVenus ΔhubP | this study | | Materials and Methods |
| Strain (Shewanella putrefaciens) | FlhF-GS-mVenus ΔfipA ΔhubP | this study | | Materials and Methods |

*Appendix 1 Continued on next page*

*Appendix 1 Continued*

| Reagent type (species) or resource | Designation | Source or reference | Identifiers | Additional information |
|---|---|---|---|---|
| Strain (Shewanella putrefaciens) | flgE1 T183C ΔflagL | *Hook et al., 2020* | | |
| Strain (Shewanella putrefaciens) | FipA-DILEL-sfGFP ΔhubP | this study | | Materials and Methods |
| Strain (Shewanella putrefaciens) | HubP-mCherry FlhF-GS-Venus | this study | | Materials and Methods |
| Strain (Shewanella putrefaciens) | FlhF-GS-mVenus ΔhubP ΔSputcn32_3157 | this study | | Materials and Methods |
| Strain (Shewanella putrefaciens) | flaB1 T166C flaA1 T174C ΔflagL FipA-DILEL-sfGFP | this study | | Materials and Methods |
| Strain (Shewanella putrefaciens) | flaB1 T166C flaA1 T174C ΔflagL ΔflhF | this study | | Materials and Methods |
| Strain (Shewanella putrefaciens) | ΔfipA fipA (Sputcn32_2550) KI | this study | | Materials and Methods |
| Strain (Shewanella putrefaciens) | FipA L118A | this study | | Materials and Methods |
| Strain (Shewanella putrefaciens) | FipA G106A | this study | | Materials and Methods |
| Strain (Shewanella putrefaciens) | FipA L118-DILEL-sfGFP | this study | | Materials and Methods |
| Strain (Shewanella putrefaciens) | FipA G106A-DILEL-sfGFP | this study | | Materials and Methods |
| Strain (Shewanella putrefaciens) | flgE1 T183C ΔflagL FliM1-GS-sfGFP | *Hook et al., 2020* | | |
| Strain (Shewanella putrefaciens) | FipA L125A | this study | | Materials and Methods |
| Strain (Shewanella putrefaciens) | flaB1 T166C flaA1 T174C ΔflagL ΔfipA ΔhubP | this study | | Materials and Methods |
| Strain (Shewanella putrefaciens) | FipA L125A-DILEL-sfGFP | this study | | Materials and Methods |
| Strain (Shewanella putrefaciens) | FlhF-mCherry FipA-DILEL-sfGFP | this study | | Materials and Methods |
| Strain (Shewanella putrefaciens) | FipA L118A FlhF-GS-mVenus | this study | | Materials and Methods |
| Strain (Shewanella putrefaciens) | FipA G106A FlhF-GS-mVenus | this study | | Materials and Methods |
| Strain (Shewanella putrefaciens) | FipA L125A FlhF-GS-mVenus | this study | | Materials and Methods |
| Strain (Shewanella putrefaciens) | FipA ΔTMD | this study | | Materials and Methods |
| Strain (Shewanella putrefaciens) | FlhF-GS-mVenus FipA ΔTMD | this study | | Materials and Methods |
| Strain (Shewanella putrefaciens) | FipA ΔTMD-DILEL-sfGFP | this study | | Materials and Methods |
| Antibody | JL-8 anti-GFP (mouse monoclonal) | Takarabio | 632380 | WB (1:10,000) |
| Antibody | Amersham ECL mouse IgG (sheep, polyclonal) | General Electric | NA931 | WB (1:5,000) |
| Antibody | Anti-GFP (mouse) | Roche | 11814460001 | WB (1:5,000) |
| Antibody | Anti-mCherry (rabbit) | Biovision | 5993–100 | WB (1:10,000) |
| Antibody | Anti-mouse | Sigma | A3562 | WB (1:5,000) |
| Antibody | Anti-rabbit | Sigma | A8025 | WB (1:20,000) |
| Recombinant DNA reagent | pDM4 | *Milton et al., 1996* | | Suicide vector for gene deletions in Vibrio sp. |
| Recombinant DNA reagent | pJH036 | *Iyer et al., 2020* | | pBAD33 derivative for sfGFP C-terminal fusion |
| Recombinant DNA reagent | pNPTS138-R6KT | *Lassak et al., 2010* | | Suicide vector for gene deletions in P. putida and S. Putrefaciens |
| Recombinant DNA reagent | pKT25 | *Karimova et al., 1998* | | For bacterial two hybrid assay |
| Recombinant DNA reagent | pKNT25 | *Karimova et al., 1998* | | For bacterial two hybrid assay |
| Recombinant DNA reagent | pUT18 | *Karimova et al., 1998* | | For bacterial two hybrid assay |
| Recombinant DNA reagent | pUT18C | *Karimova et al., 1998* | | For bacterial two hybrid assay |

*Appendix 1 Continued on next page*

*Appendix 1 Continued*

| Reagent type (species) or resource | Designation | Source or reference | Identifiers | Additional information |
|---|---|---|---|---|
| Recombinant DNA reagent | pJH003 | *Heering and Ringgaard, 2016* | | For deletion of vpa1548(lafA) |
| Recombinant DNA reagent | pSW022 | This work | | For deletion of vp2224(fipA) |
| Recombinant DNA reagent | pPM188fip | *Arroyo-Pérez and Ringgaard, 2021* | | For insertion of vp2234-sfgfp (flhF-sfgfp), replacing native flhF |
| Recombinant DNA reagent | pPM178 | this work | | For insertion of vp2224-sfgfp (fipA-sfgfp), replacing native fipA |
| Recombinant DNA reagent | pPM179 | this work | | For insertion of vp2224 (fipA) G110A point mutation in the chromosome in the native locus |
| Recombinant DNA reagent | pPM180 | this work | | For insertion of vp2224 (fipA) L129A point mutation in the chromosome in the native locus |
| Recombinant DNA reagent | pPM191 | this work | | For insertion of vp2224 (fipA) G110A fused to sfGFP in the chromosome in the native locus |
| Recombinant DNA reagent | pPM187 | this work | | For insertion of vp2224 (fipA) L129A fused to sfGFP in the chromosome in the native locus |
| Recombinant DNA reagent | pPM039 | *Arroyo-Pérez and Ringgaard, 2021* | | For deletion of vp2191 (hubP) |
| Recombinant DNA reagent | pPM194 | this work | | For overexpression of VP2224(FipA) Δ7–27 -sfGFP |
| Recombinant DNA reagent | pPM146 | this work | | For overexpression of VP2224(FipA) |
| Recombinant DNA reagent | pPM159 | this work | | For overexpression of VP2224(FipA)-sfGFP |
| Recombinant DNA reagent | pNPTS138-R6KT flhF KO (PP_4343) | this study | | For deletion of the flhF gene (PP_4343) in P. putida KT2440; Kanr |
| Recombinant DNA reagent | pNPTS138-R6KT FlhF K235A (PP_4343) | this study | | For in frame complementation of flhF (PP_4343) with FlhF K235A mutant in P. putida KT2440; Kanr |
| Recombinant DNA reagent | pNPTS138-R6KT FlhF-GS-mCherry (PP_4343) | this study | | For in frame complementation of flhF (PP_4343) with FlhF-GS-mCherry in P. putida KT2440; Kanr |
| Recombinant DNA reagent | pNPTS138-R6KT FlhF K235A-GS-mCherry (PP_4343) | this study | | For in frame complementation of flhF (PP_4343) with FlhF K235A-GS-mCherry mutant in P. putida KT2440; Kanr |
| Recombinant DNA reagent | pNPTS138-R6KT FlhF D301A-GS-mCherry (PP_4343) | this study | | For in frame complementation of flhF (PP_4343) with FlhF D301A-GS-mCherry mutant in P. putida KT2440; Kanr |
| Recombinant DNA reagent | pNPTS138-R6KT FlhF D362A-GS-mCherry (PP_4343) | this study | | For in frame complementation of flhF (PP_4343) with FlhF D362A-GS-mCherry mutant in P. putida KT2440; Kanr |
| Recombinant DNA reagent | pNPTS138-R6KT fipA KO (PP_4331) | this study | | For deletion of the fipA gene (PP_4331) in P. putida KT2440; Kanr |
| Recombinant DNA reagent | pNPTS138-R6KT fipA KI (PP_4331) | this study | | For in frame complementation of fipA (PP_4331) with wild type fipA in P. putida KT2440; Kanr |
| Recombinant DNA reagent | pNPTS138-R6KT FipA ΔTMD (AS5-22) (PP_4331) | this study | | For in frame complementation of fipA (PP_4331) with FipA ΔTMD mutant in P. putida KT2440; Kanr |

*Appendix 1 Continued on next page*

*Appendix 1 Continued*

| Reagent type (species) or resource | Designation | Source or reference | Identifiers | Additional information |
|---|---|---|---|---|
| Recombinant DNA reagent | pNPTS138-R6KT FipA G104A (PP_4331) | this study | | For in frame complementation of fipA (PP_4331) with FipA G104A mutant in P. putida KT2440; Kanr |
| Recombinant DNA reagent | pNPTS138-R6KT FipA L116A (PP_4331) | this study | | For in frame complementation of fipA (PP_4331) with FipA L116A mutant in P. putida KT2440; Kanr |
| Recombinant DNA reagent | pNPTS138-R6KT FipA L123A (PP_4331) | this study | | For in frame complementation of fipA (PP_4331) with FipA L123A mutant in P. putida KT2440; Kanr |
| Recombinant DNA reagent | pNPTS138-R6KT FipA-DILEL-sfGFP (PP_4331) | this study | | For in frame complementation of fipA (PP_4331) with FipA-DILEL-sfGFP in P. putida KT2440; Kanr |
| Recombinant DNA reagent | pNPTS138-R6KT FipA ΔTMD-DILEL-sfGFP (AS5-22) (PP_4331) | this study | | For in frame complementation of fipA (PP_4331) with FipA ΔTMD-DILEL-sfGFP mutant in P. putida KT2440; Kanr |
| Recombinant DNA reagent | pNPTS138-R6KT FipA G104A-DILEL-sfGFP (PP_4331) | this study | | For in frame complementation of fipA (PP_4331) with FipA G104A-DILEL-sfGFP mutant in P. putida KT2440; Kanr |
| Recombinant DNA reagent | pNPTS138-R6KT FipA L116A-DILEL-sfGFP (PP_4331) | this study | | For in frame complementation of fipA (PP_4331) with FipA L116A-DILEL-sfGFP mutant in P. putida KT2440; Kanr |
| Recombinant DNA reagent | pNPTS138-R6KT FipA L123A-DILEL-sfGFP (PP_4331) | this study | | For in frame complementation of fipA (PP_4331) with FipA L123A-DILEL-sfGFP mutant in P. putida KT2440; Kanr |
| Recombinant DNA reagent | pNPTS138-R6KT polar flagellar cluster KO (Sputcn32_2548–2608) | this study | | plasmid for deletion of the polar flagellar gene cluster (Sputcn32_2548–2608) in S. putrefaciens CN-32; Kanr |
| Recombinant DNA reagent | pNPTS138-R6KT lateral flagellar cluster KO (Sputcn32_3444–3485) | *Lassak et al., 2010* | | plasmid for deletion of the lateral flagellar gene cluster (Sputcn32_3444–3485) in S. putrefaciens CN-32; Kanr |
| Recombinant DNA reagent | pNPTS138-R6KT flagL KO (Sputcn32_3455, Sputcn32_3456) | *Rossmann et al., 2015* | | plasmid for deletion of the lateral flagellin genes (Sputcn32_3455, Sputcn32_3456) in S. putrefaciens CN-32; Kanr |
| Recombinant DNA reagent | pNPTS138-R6KT hubP KO (Sputcn32_2442) | *Rossmann et al., 2015* | | plasmid for deletion of the hubP gene (Sputcn32_2442) in S. putrefaciens CN-32; Kanr |
| Recombinant DNA reagent | pNPTS138-R6KT flhF KO (Sputcn32_2561) | *Rossmann et al., 2015* | | plasmid for deletion of the flhF gene (Sputcn32_2561) in S. putrefaciens CN-32; Kanr |
| Recombinant DNA reagent | pNPTS138-R6KT flhG KO (Sputcn32_2560) | *Schuhmacher et al., 2015a* | | plasmid for deletion of the flhG gene (Sputcn32_2560) in S. putrefaciens CN-32; Kanr |
| Recombinant DNA reagent | pNPTS138-R6KT FlhF-GS-Venus (Sputcn32_2561) | this study | | plasmid for in frame complementation of flhF (Sputcn32_2561) with FlhF-GS-mVenus in S. putrefaciens CN-32; Kanr |
| Recombinant DNA reagent | pNPTS138-R6KT fipA KO (Sputcn32_2550) | this study | | plasmid for deletion of the fipA gene (Sputcn32_2550) in S. putrefaciens CN-32; Kanr |
| Recombinant DNA reagent | pNPTS138-R6KT fipA KI (Sputcn32_2550) | this study | | plasmid for in frame complementation of fipA (Sputcn32_2550) with wild type fipA in S. putrefaciens CN-32; Kanr |
| Recombinant DNA reagent | pNPTS138-R6KT FipA ΔTMD (AS5-23) (Sputcn32_2550) | this study | | plasmid for in frame complementation of fipA (Sputcn32_2550) with FipA ΔTMD mutant in S. putrefaciens CN-32; Kanr |
| Recombinant DNA reagent | pNPTS138-R6KT FipA G106A (Sputcn32_2550) | this study | | plasmid for in frame complementation of fipA (Sputcn32_2550) with FipA G106A mutant in S. putrefaciens CN-32; Kanr |
| Recombinant DNA reagent | pNPTS138-R6KT FipA L118A (Sputcn32_2550) | this study | | plasmid for in frame complementation of fipA (Sputcn32_2550) with FipA L118A mutant in S. putrefaciens CN-32; Kanr |

*Appendix 1 Continued on next page*

*Appendix 1 Continued*

| Reagent type (species) or resource | Designation | Source or reference | Identifiers | Additional information |
|---|---|---|---|---|
| Recombinant DNA reagent | pNPTS138-R6KT FipA L125A (Sputcn32_2550) | this study | | plasmid for in frame complementation of fipA (Sputcn32_2550) with FipA L125 mutant in *S. putrefaciens* CN-32; Kanr |
| Recombinant DNA reagent | pNPTS138-R6KT FipA-DILEL-sfGFP (Sputcn32_2550) | this study | | plasmid for in frame complementation of fipA (Sputcn32_2550) with FipA-DILEL-sfGFP in *S. putrefaciens* CN-32; Kanr |
| Recombinant DNA reagent | pNPTS138-R6KT FipA ΔTMD-DILEL-sfGFP (AS5-23) (Sputcn32_2550) | this study | | plasmid for in frame complementation of fipA (Sputcn32_2550) with FipA ΔTMD-DILEL-sfGFP mutant in *S. putrefaciens* CN-32; Kanr |
| Recombinant DNA reagent | pNPTS138-R6KT FipA G106A-DILEL-sfGFP (Sputcn32_2550) | this study | | plasmid for in frame complementation of fipA (Sputcn32_2550) with FipA G106A-DILEL-sfGFP mutant in *S. putrefaciens* CN-32; Kanr |
| Recombinant DNA reagent | pNPTS138-R6KT FipA L116A-DILEL-sfGFP (Sputcn32_2550) | this study | | plasmid for in frame complementation of fipA (Sputcn32_2550) with FipA L116A-DILEL-sfGFP mutant in *S. putrefaciens* CN-32; Kanr |
| Recombinant DNA reagent | pNPTS138-R6KT FipA L125A-DILEL-sfGFP (Sputcn32_2550) | this study | | plasmid for in frame complementation of fipA (Sputcn32_2550) with FipA L125A mutant in *S. putrefaciens* CN-32; Kanr |
| Recombinant DNA reagent | pNPTS138-R6KT FliM1-GS-sfGFP (Sputcn32_2569) | *Hook et al., 2020* | | plasmid for in frame complementation of fliM1 (Sputcn32_2569) with FliM1-GS-sfGFP in *S. putrefaciens* CN-32; Kanr |
| Recombinant DNA reagent | pSW74 | this study | | T25-vp2224(fipA)Δ1–27 |
| Recombinant DNA reagent | pSW119 | this study | | T18-vp2224(fipA)Δ1–27 |
| Recombinant DNA reagent | pPM118 | this study | | vp2224(fipA)Δ1–27 T18 |
| Recombinant DNA reagent | pPM119 | this study | | vp2224(fipA)Δ1–27 T25 |
| Recombinant DNA reagent | pPM124 | this study | | vp2234(flhF)-T18 |
| Recombinant DNA reagent | pPM128 | this study | | vp2234(flhF)-T25 |
| Recombinant DNA reagent | pPM132 | this study | | T18-vp2234(flhF) |
| Recombinant DNA reagent | pPM136 | this study | | T25-vp2234(flhF) |
| Recombinant DNA reagent | pPM160 | this study | | T18-vp2224(fipA) Δ1–27 G110A |
| Recombinant DNA reagent | pPM161 | this study | | T18-vp2224(fipA) Δ1–27 E126A |
| Recombinant DNA reagent | pPM162 | this study | | T18-vp2224(fipA) Δ1–27 L129A |
| Recombinant DNA reagent | pKT25 FlhF (PP_4343) | this study | | plasmid for BACTH assay carrying T25-FlhF (PP_4343); Kanr |
| Recombinant DNA reagent | pKNT25 FlhF (PP_4343) | this study | | plasmid for BACTH assay carrying FlhF-T25 (PP_4343); Kanr |
| Recombinant DNA reagent | pUT18 FlhF (PP_4343) | this study | | plasmid for BACTH assay carrying FlhF-T18 (PP_4343); Ampr |
| Recombinant DNA reagent | pUT18C FlhF (PP_4343) | this study | | plasmid for BACTH assay carrying T18-FlhF (PP_4343); Ampr |
| Recombinant DNA reagent | pKT25 FlhF K235A (PP_4343) | this study | | plasmid for BACTH assay carrying T25-FlhF K235A (PP_4343); Kanr |
| Recombinant DNA reagent | pKNT25 FlhF K235A (PP_4343) | this study | | plasmid for BACTH assay carrying FlhF K235A -T25 (PP_4343); Kanr |

*Appendix 1 Continued on next page*

*Appendix 1 Continued*

| Reagent type (species) or resource | Designation | Source or reference | Identifiers | Additional information |
|---|---|---|---|---|
| Recombinant DNA reagent | pUT18 FlhF K235A (PP_4343) | this study | | plasmid for BACTH assay carrying FlhF K235A -T18 (PP_4343); Ampr |
| Recombinant DNA reagent | pUT18C FlhF K235A (PP_4343) | this study | | plasmid for BACTH assay carrying T18-FlhF K235A (PP_4343); Ampr |
| Recombinant DNA reagent | pKT25 FipA (PP_4331) | this study | | plasmid for BACTH assay carrying T25-FipA (PP_4331); Kanr |
| Recombinant DNA reagent | pKNT25 FipA (PP_4331) | this study | | plasmid for BACTH assay carrying FipA-T25 (PP_4331); Kanr |
| Recombinant DNA reagent | pUT18 FipA (PP_4331) | this study | | plasmid for BACTH assay carrying FipA-T18 (PP_4331); Ampr |
| Recombinant DNA reagent | pUT18C FipA (PP_4331) | this study | | plasmid for BACTH assay carrying T18-FipA (PP_4331); Ampr |
| Recombinant DNA reagent | pKT25 FipA G104A (PP_4331) | this study | | plasmid for BACTH assay carrying T25-FipA G104A (PP_4331); Kanr |
| Recombinant DNA reagent | pKNT25 FipA G104A (PP_4331) | this study | | plasmid for BACTH assay carrying FipA G104A -T25 (PP_4331); Kanr |
| Recombinant DNA reagent | pUT18 FipA G104A (PP_4331) | this study | | plasmid for BACTH assay carrying FipA G104A -T18 (PP_4331); Ampr |
| Recombinant DNA reagent | pUT18C FipA G104A (PP_4331) | this study | | plasmid for BACTH assay carrying T18-FipA G104A (PP_4331); Ampr |
| Recombinant DNA reagent | pKT25 FipA L116A (PP_4331) | this study | | plasmid for BACTH assay carrying T25-FipA L116A (PP_4331); Kanr |
| Recombinant DNA reagent | pKNT25 FipA L116A (PP_4331) | this study | | plasmid for BACTH assay carrying FipA L116A -T25 (PP_4331); Kanr |
| Recombinant DNA reagent | pUT18 FipA L116A (PP_4331) | this study | | plasmid for BACTH assay carrying FipA L116A -T18 (PP_4331); Ampr |
| Recombinant DNA reagent | pUT18C FipA L116A (PP_4331) | this study | | plasmid for BACTH assay carrying T18-FipA L116A (PP_4331); Ampr |
| Recombinant DNA reagent | pKT25 FipA L125A (PP_4331) | this study | | plasmid for BACTH assay carrying T25-FipA L123A (PP_4331); Kanr |
| Recombinant DNA reagent | pKNT25 FipA L125A (PP_4331) | this study | | plasmid for BACTH assay carrying FipA L123A -T25 (PP_4331); Kanr |
| Recombinant DNA reagent | pUT18 FipA L125A (PP_4331) | this study | | plasmid for BACTH assay carrying FipA L123A -T18 (PP_4331); Ampr |
| Recombinant DNA reagent | pUT18C FipA L125A (PP_4331) | this study | | plasmid for BACTH assay carrying T18-FipA L123A (PP_4331); Ampr |
| Recombinant DNA reagent | pKT25 FlhF (Sputcn32_2561) | this study | | plasmid for BACTH assay carrying T25-FlhF (Sputcn32_2561); Kanr |
| Recombinant DNA reagent | pKNT25 FlhF (Sputcn32_2561) | this study | | plasmid for BACTH assay carrying FlhF-T25 (Sputcn32_2561); Kanr |

*Appendix 1 Continued*

| Reagent type (species) or resource | Designation | Source or reference | Identifiers | Additional information |
|---|---|---|---|---|
| Recombinant DNA reagent | pUT18 FlhF (Sputcn32_2561) | this study | | plasmid for BACTH assay carrying FlhF-T18 (Sputcn32_2561); Ampr |
| Recombinant DNA reagent | pUT18C FlhF (Sputcn32_2561) | this study | | plasmid for BACTH assay carrying T18-FlhF (Sputcn32_2561); Ampr |
| Recombinant DNA reagent | pKT25 FipA (Sputcn32_2550) | this study | | plasmid for BACTH assay carrying T25-FipA (Sputcn32_2550); Kanr |
| Recombinant DNA reagent | pKNT25 FipA (Sputcn32_2550) | this study | | plasmid for BACTH assay carrying FipA-T25 (Sputcn32_2550); Kanr |
| Recombinant DNA reagent | pUT18 FipA (Sputcn32_2550) | this study | | plasmid for BACTH assay carrying FipA-T18 (Sputcn32_2550); Ampr |
| Recombinant DNA reagent | pUT18C FipA (Sputcn32_2550) | this study | | plasmid for BACTH assay carrying T18-FipA (Sputcn32_2550); Ampr |
| Recombinant DNA reagent | pKT25 FipA G106A (Sputcn32_2550) | this study | | plasmid for BACTH assay carrying T25-FipA G106A (Sputcn32_2550); Kanr |
| Recombinant DNA reagent | pKNT25 FipA G106A (Sputcn32_2550) | this study | | plasmid for BACTH assay carrying FipA G106A -T25 (Sputcn32_2550); Kanr |
| Recombinant DNA reagent | pUT18 FipA G106A (Sputcn32_2550) | this study | | plasmid for BACTH assay carrying FipA G106A -T18 (Sputcn32_2550); Ampr |
| Recombinant DNA reagent | pUT18C FipA G106A (Sputcn32_2550) | this study | | plasmid for BACTH assay carrying T18-FipA G106A (Sputcn32_2550); Ampr |
| Recombinant DNA reagent | pKT25 FipA L116A (Sputcn32_2550) | this study | | plasmid for BACTH assay carrying T25-FipA L116A (Sputcn32_2550); Kanr |
| Recombinant DNA reagent | pKNT25 FipA L116A (Sputcn32_2550) | this study | | plasmid for BACTH assay carrying FipA L116A -T25 (Sputcn32_2550); Kanr |
| Recombinant DNA reagent | pUT18 FipA L116A (Sputcn32_2550) | this study | | plasmid for BACTH assay carrying FipA L116A -T18 (Sputcn32_2550); Ampr |
| Recombinant DNA reagent | pUT18C FipA L116A (Sputcn32_2550) | this study | | plasmid for BACTH assay carrying T18-FipA L116A (Sputcn32_2550); Ampr |
| Recombinant DNA reagent | pKT25 FipA L125A (Sputcn32_2550) | this study | | plasmid for BACTH assay carrying T25-FipA L125A (Sputcn32_2550); Kanr |
| Recombinant DNA reagent | pKNT25 FipA L125A (Sputcn32_2550) | this study | | plasmid for BACTH assay carrying FipA L125A -T25 (Sputcn32_2550); Kanr |
| Recombinant DNA reagent | pUT18 FipA L125A (Sputcn32_2550) | this study | | plasmid for BACTH assay carrying FipA L125A -T18 (Sputcn32_2550); Ampr |
| Recombinant DNA reagent | pUT18C FipA L125A (Sputcn32_2550) | this study | | plasmid for BACTH assay carrying T18-FipA L125A (Sputcn32_2550); Ampr |
| Sequence-based reagent | VP2224-del-a | this study | PCR primers | CCCCC tctaga ACGTTGT CATGCTTGGTGAAAGCA |
| Sequence-based reagent | VP2224-del-b | this study | PCR primers | AGTCTCTTCAGCCAT CGTCATTC |
| Sequence-based reagent | VP2224-del-c | this study | PCR primers | gaatgacgatggctgaagagact cgacgataaagagaataaaaagaagc |

*Appendix 1 Continued on next page*

*Appendix 1 Continued*

| Reagent type (species) or resource | Designation | Source or reference | Identifiers | Additional information |
|---|---|---|---|---|
| Sequence-based reagent | VP2224-del-d | this study | PCR primers | CCCCC tctaga ACGCGACGCTGCTGA CCCGCAGAA |
| Sequence-based reagent | VP2224-check | this study | PCR primers | acaaactccgtggggatgaatac |
| Sequence-based reagent | vp2224 AA1-6/28-end w/o Stop | this study | PCR primers | ccccc ctcaga atg gctgaagagacttttctgcgc |
| Sequence-based reagent | pUT18C/pKT25-vp2234-cw | this study | PCR primers | ccccc tctaga G aaaataaagcgatttttttgccaaagac |
| Sequence-based reagent | pUT18C/pKT25-vp2234-ccw | this study | PCR primers | ccccc ggtacc ctagagtccttcgttgtcactg |
| Sequence-based reagent | vpa1548-del-d | this study | PCR primers | Ccccc ctcgag TTATGTGTTCCGCC TTCCTCTC |
| Sequence-based reagent | vpa1548-del-chk | this study | PCR primers | aagtagccacatcccaaacgc |
| Sequence-based reagent | VP2191-del-d | this study | PCR primers | ccccc tctaga GACAATGCGCT GCACGGAAT |
| Sequence-based reagent | VP2191-del-chk | this study | PCR primers | gatggaaaacggctacacca |
| Sequence-based reagent | del vp2234(FlhF)-d | this study | PCR primers | CCCCC tctaga GAATACATGCTAC GAGCTCAAGG |
| Sequence-based reagent | del vp2234(FlhF)-chk | this study | PCR primers | GTTTACGGCATG ATTGATGGCG |
| Sequence-based reagent | vp2224-Gly110Ala-cw | this study | PCR primers | gagcaaccaaaatggtgcagttaGCGg ctgatatcaacgagctaatcg |
| Sequence-based reagent | vp2224-Gly110Ala-ccw | this study | PCR primers | CGATTAGCTCGTTGATATCAGCcgcT AACTGCACCATTTTGGTTGCTC |
| Sequence-based reagent | vp2224-Glu126Ala-cw | this study | PCR primers | agagtgtgaactgccaaaagcaGCAgc agagttgatgctctctttgc |
| Sequence-based reagent | vp2224-Glu126Ala-ccw | this study | PCR primers | GCAAAGAGAGCA TCAACTCTGctg CTGCTTTTGGCAGT TCACACTCT |
| Sequence-based reagent | vp2224-Leu129Ala-cw | this study | PCR primers | tgaactgccaaaagcagaagcagag GC gatgctctc tttgcagaaaaactg |
| Sequence-based reagent | vp2224-Leu129Ala-ccw | this study | PCR primers | CAG TTT TTT CTG CAA AGA GAG CAT CGC CTC TGC TTC TGC TTT TGG CAG TTC A |
| Sequence-based reagent | C-term sfGFP-vp2224-a | this study | PCR primers | CCCCC actagt ATGGCTGAAG AGACTTTTTATCTGTAC |
| Sequence-based reagent | C-term sfGFP-vp2224-b | this study | PCR primers | gagctcgaggatgtc TCGTCGACGCCCACGTGG |
| Sequence-based reagent | C-term sfGFP-vp2224-c | this study | PCR primers | gacatcctcgagctc atgagcaaaggagaagaactttcac |
| Sequence-based reagent | C-term sfGFP-vp2224-d | this study | PCR primers | tta tttgtagagctcatccatgcc |
| Sequence-based reagent | C-term sfGFP-vp2224-e | this study | PCR primers | ggcatggatgagctctacaaa taa AGAGAATAAAAG AAGCTTCGG |
| Sequence-based reagent | C-term sfGFP-vp2224-f | this study | PCR primers | ccccc gcatgc TTTGTTT GTCGATTGCTGTTAGTGG |
| Sequence-based reagent | del AA7-27 vp2224-b | this study | PCR primers | AAAAGTCTCTTCAGCCATCGTCATTC |
| Sequence-based reagent | del AA7-27 vp2224-c | this study | PCR primers | GAATGACGATGGCTGAAGAGACTTTT CTGCGCATTCGTGCTAGTTTGC |
| Sequence-based reagent | vp2224-cw-pBAD | this study | PCR primers | CCCCC tctaga atggctga agagactttttatctg |

*Appendix 1 Continued on next page*

*Appendix 1 Continued*

| Reagent type (species) or resource | Designation | Source or reference | Identifiers | Additional information |
|---|---|---|---|---|
| Sequence-based reagent | vp2224-ccw-pBAD | this study | PCR primers | CCCCC gcatgc tta tcgtcgacgcccacg |
| Sequence-based reagent | vp2224 cw restore deletion | this study | PCR primers | ACCTATAATTGGCTGAATGACG ATGGCTGAAGAGACTT TTTTATCTGTAC |
| Sequence-based reagent | downstream vp2224 cw | this study | PCR primers | AGAGAATAAAAAGAAGCTTCGGC |
| Sequence-based reagent | pUT18/pKNT25- vp2224-cw | this study | PCR primers | ccccc TCTAGA atgg ctgaagagactttttatctgtac |
| Sequence-based reagent | pUT18/pKNT25- tr-vp2224-cw | this study | PCR primers | ccccc TCTAGA ATG cgcattcgtgctagtttgc |
| Sequence-based reagent | pUT18/pKNT25-vp2222 -ccw | this study | PCR primers | ccccc GGTACC CG tcgtcgacgcccacgtg |
| Sequence-based reagent | pUT18C/pKT25-vp2224-cw | this study | PCR primers | ccccc tctaga G gctgaa gagactttttatctgtac |
| Sequence-based reagent | pUT18C/pKT25-vp2224-ccw | this study | PCR primers | ccccc ggtacc ttatc gtcgacgcccacgtg |
| Sequence-based reagent | tr2224 put18C cw | this study | PCR primers | ccccc tctaga G ATG cgcattcgtgctagtttgcaaaa |
| Sequence-based reagent | sfGFP-1-ccw | this study | PCR primers | ccccc tctaga tttgtagagc tcatccatgccatg |
| Sequence-based reagent | vp2224 C-term PhoA-LacZ cw | this study | PCR primers | CCCCC tctaga g atggcccggacaccagaaatg |
| Sequence-based reagent | end -LacZ w/o STOP ccw | this study | PCR primers | gcgccattcgccattcaggctgc |
| Sequence-based reagent | LacZ to vp2224 w/o ATG | this study | PCR primers | CCT GAA TGG CGA ATG GCG C GCT GAA GAG ACT TTT TTA TCT GTA CC |
| Sequence-based reagent | end vp2224 ccw | this study | PCR primers | CCCCC aagctt ttatcgtcgacgcccacgtgg |
| Sequence-based reagent | vp2224 ccw restore deletion | this study | PCR primers | GCCGAAGCTTCTTTTTATTCTCT TTATC GTCGACGCCCACGTG |
| Sequence-based reagent | M13 | this study | PCR primers | TGTAAAACGACGGCCAGTCC |
| Sequence-based reagent | M13r | this study | PCR primers | CACACAGGAAACAGCTATGACC |
| Sequence-based reagent | flhF1-flhG1 fwd | this study | PCR primers | GCGCTGAGTGTGTTGATCCAAA |
| Sequence-based reagent | EcoRV FliM1 N-term fwd | this study | PCR primers | GCGAATTCGTGGATCCA GATGCTCATTGAAGA TGCTCTCCTG |
| Sequence-based reagent | EcoRV FliM1 N-term rev | this study | PCR primers | GCCAAGCTTCTCTGCAG GATAATAAAACTGCG GCCCACTTCC |
| Sequence-based reagent | Check-GFP FliM1-fwd | this study | PCR primers | GCAGTTCAGATGAGTCATCCTC |
| Sequence-based reagent | Check-GFP FliM1 KO-rev | this study | PCR primers | GACATTTTGGCAGTTGATGCGAC |
| Sequence-based reagent | OL FliM1 GFP rev | this study | PCR primers | GAAAAGTTCTTCTCCTT TGCTGCTGCCTAATTCA GATATATCTCTAGCTTTGCCTTTGC |
| Sequence-based reagent | OL FliM1 GFP fwd | this study | PCR primers | GGATGAGCTCTACAAAGG ATCCTAAGGTGAAGCAAG ATGAGCACAGAAGATA |
| Sequence-based reagent | EcoRV FlhF C-term fwd | this study | PCR primers | GCGAATTCGTGGATCC AGATGCAAGAAATGGT TGGACAGCCT |
| Sequence-based reagent | EcoRV FlhF C-term rev | this study | PCR primers | GCCAAGCTTCTCTGC AGGATGCCACATCTAAA AATCGGTCGG |
| Sequence-based reagent | Check-FlhF-FLAG-fwd | this study | PCR primers | GCATCAGTCAATGCAAGCAACC |
| Sequence-based reagent | OL-FlhF-Venus rev | this study | PCR primers | CACGCTGCCCTCAAAT GCACAGGCCATATTATCTG |

*Appendix 1 Continued on next page*

*Appendix 1 Continued*

| Reagent type (species) or resource | Designation | Source or reference | Identifiers | Additional information |
|---|---|---|---|---|
| Sequence-based reagent | OL_Venus fwd | this study | PCR primers | GCATTTGAGGGCAG CGTGAGCAAGGGCGAG GAGCTGTT |
| Sequence-based reagent | OL_Venus rev | this study | PCR primers | GTCATAACTTTACTTGTA CAGCTCGTCCATGCC |
| Sequence-based reagent | OL-FlhF-Venus fwd | this study | PCR primers | TACAAGTAAAGTTATGAC CCTGGATCAAGCAAG |
| Sequence-based reagent | FlhF-Ven Seq_Primer | this study | PCR primers | GCTGAGTTAGTACGAGCACTAC |
| Sequence-based reagent | FlhG-Ven Seq_Primer | this study | PCR primers | CGATATTATTGTCCGTGGGCCT |
| sequence-based reagent | FlhF-Ven Seq_Primer fwd | this study | PCR primers | GCTGTTGTAGTTGTACTCCAGC |
| sequence-based reagent | EcoRV FlhF C-term rev | this study | PCR primers | GCCAAGCTTCTCTGCAGGA TGCCACATCTAAAAATCGGTCGG |
| sequence-based reagent | EcoRV-2550-GFP-fwd | this study | PCR primers | GCGAATTCGTGGATCCAGA TGCCATCAATAACGGAAAAGGGG |
| sequence-based reagent | OL-2550-GFP-rev | this study | PCR primers | GAAAAGTTCTTCTCCTTT GCTCAGTTCCAGAATATCT TTACGATGTAACCGGATC AATAATTCAGC |
| sequence-based reagent | OL-2550-GFP-fwd | this study | PCR primers | GGATGAGCTCTACAAAG GATCCTAACGAAGTGT AGGGGCTAAGACG |
| sequence-based reagent | EcoRV-2550-GFP-rev | this study | PCR primers | GCCAAGCTTCTCTGCAG GATGCCTTTGTTTATAT GCTCGACGG |
| sequence-based reagent | Check-2550-GFP-fwd | this study | PCR primers | CGATGAAGAATGGGCTGAACTC |
| sequence-based reagent | Check-2550-GFP-rev | this study | PCR primers | CGAAGGATGCGAGAATGACGAA |
| sequence-based reagent | OL-2069_FlhF-rev | this study | PCR primers | AATCTTCACTAGCAT CCCCGTACATTGAACTC |
| sequence-based reagent | OL-FlhF-Ven-fwd | this study | PCR primers | GGGATGCTAGTGAAGA TTAAACGATTTTTTGCCAAAGAC |
| sequence-based reagent | OL-FlhF-Ven-rev | this study | PCR primers | AACATTAGCTTACTTGT ACAGCTCGTCCATGC |
| sequence-based reagent | OL-2068-fwd | this study | PCR primers | TACAAGTAAGCTAATGT TTTAGGGTCTTACGCG |
| sequence-based reagent | BACTH 2550 pkT25 fwd | this study | PCR primers | CAGGGTCGACTCTAGA GGGCGATGAATTTTTGATCGCGG |
| sequence-based reagent | BACTH 2550 pkT25 rev | this study | PCR primers | TTAGTTACTTAGGTAC CCGGGGTTTACGATG TAACCGGATCAATAATTCAGC |
| sequence-based reagent | BACTH 2550 fwd | this study | PCR primers | CTGCAGGTCGACTCTAGA GGGCGATGAATTTTTGATCGCGG |
| Sequence-based reagent | BACTH 2550 rev | this study | PCR primers | GAGCTCGGTACCCGG GGTTTACGATGTAACC GGATCAATAATTCAGC |
| Sequence-based reagent | OL_FliM1 mCh rev | this study | PCR primers | TTTGTATAACTCATCCATACCA |
| Sequence-based reagent | FlhF-Ven Seq_Primer rev | this study | PCR primers | GCTGGAGTACAACTACAACAGC |
| Sequence-based reagent | OL-GFP-fwd | this study | PCR primers | AGCAAAGGAGAAGAACTTTTC |
| Sequence-based reagent | OL-GFP-rev | this study | PCR primers | GGATCCTTTGTAGAGCTCATCC |
| Sequence-based reagent | OL -mCherry fwd | this study | PCR primers | GTTTCCAAAGGGGAAGAGGACA |
| Sequence-based reagent | pKT25-for | this study | PCR primers | CACTGACGGCGGAT ATCGACATGTT |
| Sequence-based reagent | pKT25-rev | this study | PCR primers | CCGCCGGACATC AGCGCCATTC |
| Sequence-based reagent | pUT18-for | this study | PCR primers | CCAGGCTTTACACTTTATGCTTCC |

*Appendix 1 Continued on next page*

*Appendix 1 Continued*

| Reagent type (species) or resource | Designation | Source or reference | Identifiers | Additional information |
|---|---|---|---|---|
| Sequence-based reagent | pUT18-rev | this study | PCR primers | GACGCGCCTCGGTGCCCACTGC |
| Sequence-based reagent | pKNT25-for | this study | PCR primers | CCCAGGCTTTACACTTTATGCTTCC |
| Sequence-based reagent | pKNT25-rev | this study | PCR primers | GTTTTTTTCCTTCGCCACGGCCTTG |
| Sequence-based reagent | pUT18C-for | this study | PCR primers | CGGCGTGCCGAGCGGACGTTCG |
| Sequence-based reagent | pUT18C-rev | this study | PCR primers | TCAGCGGGTGTTGGCGGGTGTC |
| Sequence-based reagent | FlhF Seq_Primer fwd | this study | PCR primers | GCCCACTTTGGATCAACACACT |
| Sequence-based reagent | FlhF Seq_Primer rev | this study | PCR primers | CGTGCTCACAAAACTCGATGAA |
| Sequence-based reagent | EcoRV FliFG1 KO fwd | this study | PCR primers | GCGAATTCGTGGATCC AGATGCCGAAAACTTG TGGCTGAAAA |
| Sequence-based reagent | OL- FliFG1 KO rev | this study | PCR primers | ATCGCCACCCCCGAC AATCATTTCTGTGCTC |
| Sequence-based reagent | OL- FliFG1 KO fwd | this study | PCR primers | ATTGTCGGGGGTGGCGA TGAGTTCCTCTAAT |
| Sequence-based reagent | EcoRV FliFG1 KO rev | this study | PCR primers | GCCAAGCTTCTCTGC AGGATGCAACCTAATA GTCACTGCTTG |
| Sequence-based reagent | OL-fipA L118A rev | this study | PCR primers | AGCTTCAGCTTTGG GCGCTTCACA |
| Sequence-based reagent | OL-fipA L118A fwd | this study | PCR primers | ATAAAAGAGTGTGA AGCGCCCAAA |
| Sequence-based reagent | OL-fipA G106A rev | this study | PCR primers | TTCATCGACTCCCGCGGCAAGTCC |
| Sequence-based reagent | OL-fipA G106A fwd | this study | PCR primers | AAAATGGTCGGACT TGCCGCGGGA |
| Sequence-based reagent | OL-PPfipA G104A rev | this study | PCR primers | CATCGATACTCGCA GCCATCCC |
| Sequence-based reagent | OL-PPfipA G104A fwd | this study | PCR primers | GCTGGTGGGGA TGGCTGCGAGT |
| Sequence-based reagent | OL-PPfipA L123A rev | this study | PCR primers | ACACCTTGCTCATCGCCTCCGC |
| Sequence-based reagent | OL-PPfipA L123A fwd | this study | PCR primers | GGCCGAGGCGGAGGCGATGAGC |
| Sequence-based reagent | EcoRV-flhF KO-fwd | this study | PCR primers | GCCAAGCTTCTCTGC AGGATGCATAGGCGT CGGTGATTGAGG |
| Sequence-based reagent | OL-flhF KO-rev | this study | PCR primers | TAAGTGAAGGCATTTGA GTAGAGTTATGACCCTGG |
| Sequence-based reagent | OL-flhF KO-fwd | this study | PCR primers | CTCAAATGCCTTCAC TATGCGTCCTCTACTGG |
| Sequence-based reagent | EcoRV-flhF KO-rev | this study | PCR primers | GCGAATTCGTGGATCC AGATGCTAAGCATTCTC CTAAGCTTGTTG |
| Sequence-based reagent | OL-fipA L125A rev | this study | PCR primers | TAACCGGATCAAGG CTTCAGCTTC |
| Sequence-based reagent | OL-fipA L125A fwd | this study | PCR primers | GCTGAAGCTGAAGCC TTGATCCGG |
| Sequence-based reagent | EcoRV FlhF sub rev | this study | PCR primers | GCCAAGCTTCTCTGC AGGATGCTCGTCACA TACAACGACTAG |
| Sequence-based reagent | BACTH 2550 L125A pkT25 rev | this study | PCR primers | TTAGTTACTTAGGTACCC GGGGTTTACGATGTAAC CGGATCAAGGCTTCAGC |
| Sequence-based reagent | BACTH 2550 L125A rev | this study | PCR primers | GAGCTCGGTACCCGG GGTTTACGATGTAACC GGATCAAGGCTTCAGC |

*Appendix 1 Continued on next page*

*Appendix 1 Continued*

| Reagent type (species) or resource | Designation | Source or reference | Identifiers | Additional information |
|---|---|---|---|---|
| Sequence-based reagent | OL-FipA L125A-GFP-rev | this study | PCR primers | GAAAAGTTCTTCTCCT TTGCTCAGTTCCAGA ATATCTTTACGATGTAA CCGGATCAAGGCTTCAGC |
| Sequence-based reagent | BACTH FlhF pkT25 fwd | this study | PCR primers | CAGGGTCGACTCTAGAG AAGATTAAACGATTTTTT GCCAAAGACA |
| Sequence-based reagent | BACTH FlhF pkT25 rev | this study | PCR primers | TTAGTTACTTAGGTACC CGGGGCTCAAATGC ACAGGCCATATTATCT |
| Sequence-based reagent | BACTH FlhF fwd | this study | PCR primers | CTGCAGGTCGACTCTAG AGAAGATTAAACGAT TTTTTGCCAAAGACA |
| Sequence-based reagent | BACTH FlhF rev | this study | PCR primers | GAGCTCGGTACCCG GGGCTCAAATGCACA GGCCATATTATCT |
| Sequence-based reagent | BACTH FlhF GTG fwd | this study | PCR primers | CTGCAGGTCGACTCT AGAGGTGAAGATTAA ACGATTTTTTGCCAAAG |
| Sequence-based reagent | EcoRV_FipA KO fwd | this study | PCR primers | GCGAATTCGTGGATCCAGA TTTTTAGGTATCATTAAC TTACGTGGTAATGT |
| Sequence-based reagent | OL-FipA KO rev | this study | PCR primers | ACACTTCGCTATTTACGATG ATCGCCCATTAAAAATCCTTATGCA |
| Sequence-based reagent | OL-FipA KO fwd | this study | PCR primers | AAGGATTTTTAATGGGC GATCATCGTAAATAGCGAA GTGTAGGG |
| Sequence-based reagent | EcoRV-FipA KO rev | this study | PCR primers | GCCAAGCTTCTCTGCAGG ATGAACTGATCGCCT TTGTTTATATGC |
| Sequence-based reagent | Check-FipA KO fwd | this study | PCR primers | AAGAAATGTCGCAGCCGTAGC |
| Sequence-based reagent | Check-FipA KO rev | this study | PCR primers | CCAGTTGCGACAATCTTCGGAG |
| Sequence-based reagent | OL-PPfipA L116A rev | this study | PCR primers | CATCAACTCCGCCTCGGCC TGGGTCGCGCCGCAGCTCTGGGT |
| Sequence-based reagent | OL-PPfipA L1164A fwd | this study | PCR primers | GAGTTGACCCAGAGCTGCGG CGCGACCCAGGCCGAGGCG |
| Sequence-based reagent | Check-PP_4331 (FipA) fwd | this study | PCR primers | GCTTACGAACAGAACGCAAGGC |
| Sequence-based reagent | Check-PP_4331 (FipA) rev | this study | PCR primers | GCAATACGTGATTCGGTGCAG |
| Sequence-based reagent | EcoRV-PP_4331 KO-fwd | this study | PCR primers | GCGAATTCGTGGATCCAGATGC AGATGCACGCCAAACAGAAA |
| Sequence-based reagent | PP_4331 KO-OL-rev | this study | PCR primers | TCAAGGAGCTAGGATCAACTC AGATGTTCTCCAGC |
| Sequence-based reagent | PP_4331KO-OL-fwd | this study | PCR primers | TTGATCCTAGCTCCTTGAC GGGGTACCCTCG |
| Sequence-based reagent | EcoRV-PP_4331 KO-rev | this study | PCR primers | GCCAAGCTTCTCTGCAGGA TGCATGAATTGCCTGTA CAACACCA |
| Sequence-based reagent | Check-PP_4331KO-fwd | this study | PCR primers | GCGAAACGATCGATCAGGTCGA |
| Sequence-based reagent | Check-PP_4331KO-rev | this study | PCR primers | GCACCGTAATCGAACACATGTG |
| Sequence-based reagent | EcoRV-PP_4331-GFP-fwd | this study | PCR primers | GCGAATTCGTGGATCCAGA TGCAGATGCACGCC AAACAGAAA |
| Sequence-based reagent | PP_4331-GFP-OL-rev | this study | PCR primers | GAAAAGTTCTTCTCCTTTGCTCA GTTCCAGAATATCAGGA GCCCGGTACACCTTGCTC |
| Sequence-based reagent | PP_43310-GFP-OL-fwd | this study | PCR primers | GGATGAGCTCTACAAAGGATCCT GACGGGGTACCCTCGGCAGCA |

*Appendix 1 Continued on next page*

| Reagent type (species) or resource | Designation | Source or reference | Identifiers | Additional information |
|---|---|---|---|---|
| Sequence-based reagent | EcoRV-PP_4331-GFP-rev | this study | PCR primers | GCCAAGCTTCTCTGCAGGATGC ATGAATTGCCTGTACAACACCA |
| Sequence-based reagent | EcoRV-FlhF-mCh-fwd | this study | PCR primers | GCGAATTCGTGGATCCAGATGCA TGGACAGCTTCCGTATCGG |
| Sequence-based reagent | FlhF-mCh-OL-rev | this study | PCR primers | CTCTTCCCCTTTGGAAACGCTGCC ACCCGCTCGCCGTGGGTTGTGA |
| Sequence-based reagent | FlhF-mCh-OL-fwd | this study | PCR primers | ATGGATGAGTTATACAAATGACCAT GAAGCGTGTGCAAAG |
| Sequence-based reagent | EcoRV-FlhF-mCh-rev | this study | PCR primers | GCCAAGCTTCTCTGCAGGATG CCAACACACGGAAACGGTTCA |
| Sequence-based reagent | Check-PP_4343 KO-fwd | this study | PCR primers | GCCTGAAATCGAGCCGATCGAA |
| Sequence-based reagent | Check-PP_4343 KO-rev | this study | PCR primers | GCGTCGGTAATCGAGGTAGGTT |
| Sequence-based reagent | BACTH PP FipA pkT25 fwd | this study | PCR primers | CAGGGTCGACTCTAGAGATC CTAGAGGTTGCTGTCATCT |
| Sequence-based reagent | BACTH PP FipA pkT25 rev | this study | PCR primers | TTAGTTACTTAGGTACCCGGG GAGGAGCCCGGTACACCTTGCTC |
| Sequence-based reagent | BACTH PP FipA fwd | this study | PCR primers | CTGCAGGTCGACTCTAGAGAT CCTAGAGGTTGCTGTCATCT |
| Sequence-based reagent | BACTH PP FipA rev | this study | PCR primers | GAGCTCGGTACCCGGGGAGGA GCCCGGTACACCTTGCTC |
| Sequence-based reagent | BACTH PP FlhF pkT25 fwd | this study | PCR primers | CAGGGTCGACTCTAGAGCAAGT TAAGCGATTTTTCGCCGC |
| Sequence-based reagent | BACTH PP FlhF pkT25 rev | this study | PCR primers | TTAGTTACTTAGGTACCCGGGGAC CCGCTCGCCGTGGGTTGTGA |
| Sequence-based reagent | BACTH PP FlhF fwd | this study | PCR primers | CTGCAGGTCGACTCTAGAGCAAG TTAAGCGATTTTTCGCCGC |
| Sequence-based reagent | BACTH PP FlhF rev | this study | PCR primers | GAGCTCGGTACCCGGGGACC CGCTCGCCGTGGGTTGTGA |
| Sequence-based reagent | EcoRV FlhF sub fwd | this study | PCR primers | GCGAATTCGTGGATCCAGA TGCATCAGTCAATGCAAGCAACC |
| Sequence-based reagent | Check-FlhF KI/O-fwd | this study | PCR primers | GCCACTGGGTAGTGTCGTAAAA |
| Sequence-based reagent | OL-FlhF D328A rev | this study | PCR primers | CCCCATACCAGCGGTGGCTATCAATAC |
| Sequence-based reagent | OL-FlhF D328A fwd | this study | PCR primers | AAGCTAGTATTGATAGCCACCGCTGGT |
| Sequence-based reagent | EcoRV PPFlhF sub fwd | this study | PCR primers | GCGAATTCGTGGATCCAGAT GCATGTTCTGGCGTATCAGGAA |
| Sequence-based reagent | OL-FlhF K235A rev | this study | PCR primers | GCGCGCGGCCAGCGCGGCCAGGGT |
| Sequence-based reagent | OL-FlhF K235A fwd | this study | PCR primers | GGCAAGACCACCACCCTGGC CGCGCTGGCCGCG |
| Sequence-based reagent | EcoRV PPFlhF sub rev | this study | PCR primers | GCCAAGCTTCTCTGCAGGATGCAT GCTACCCATGTCTGTTCT |
| Sequence-based reagent | Check-PP_4343 KI-fwd | this study | PCR primers | GCTACCAGTGATTACCCTGGAG |
| Sequence-based reagent | EcoRV PPFlhF sub 1 rev | this study | PCR primers | GCCAAGCTTCTCTGCAGGATG CGTCGGTAATCGAGGTAGGTT |
| Sequence-based reagent | KT2440 FlhF Seq primer rev | this study | PCR primers | GCTGGTGAGCATGGACAGCTTC |
| Sequence-based reagent | EcoRV-FipA dTM fwd | this study | PCR primers | GCGAATTCGTGGATCCAGATG CCGTAGCTGCAAGTAAAGATG |
| Sequence-based reagent | OL-FipA dTM rev | this study | PCR primers | CTGCTTTTGTTCATCGCC CATTAAAAATCCTTATGC |
| Sequence-based reagent | OL-FipA dTM fwd | this study | PCR primers | GGCGATGAACAAAAGCAGTT GAGTAAATTACGTAATAAAGTTG |
| Sequence-based reagent | OL-PP_FipA dTM rev | this study | PCR primers | GCTGTAGTTCTCTAGGAT CAACTCAGATGTTCTCC |
| Sequence-based reagent | OL-PP_FipA dTM fwd | this study | PCR primers | ATCCTAGAGAACTACAGCAAG CGCCAGCGCG |
| Software, algorithm | cellProfiles (R package) | *Cameron et al., 2014*; *Cameron, 2018* | https://github.com/ta-cameron/Cell-Profiles | |

| Reagent type (species) or resource | Designation | Source or reference | Identifiers | Additional information |
|---|---|---|---|---|
| Software, algorithm | MicrobeJ | *Ducret et al., 2016* | | |

