## [Editor Report · eLife assessment]

This **important** study describes the discovery of a mechanism by which multiple species of bacteria synthesize and localize polar flagella via a novel protein, FipA, which interacts with FlhF. The authors use appropriate methodological approaches (biochemistry, molecular microbiology, quantitative microscopy, and bacterial genetics) to obtain and present **convincing** results and interpretations. This work will particularly interest those studying bacterial motility and bacterial cell biologists.

---

## [Referee Report · Reviewer #1 (Public review)]

Summary:

Bacteria exhibit species-specific numbers and localization patterns of flagella. How specificity in number and pattern is achieved is poorly understood but often depends on a soluble GTPase called FlhF. Here the authors take an unbiased protein-pulldown approach to identify a protein FipA in V. parahaemolyticus that interacts with FlhF. They show that FipA co-occurs with FlhF in the genomes of bacteria with polarly-localized flagella and study the role of FipA in three different bacteria: V. parahaemolyticus, S. purtefaciens, and P. putida. In each case, they show that FipA contributes to FlhF polar localization, flagellar assembly, flagellar patterning, and motility to different species-specific extents.

Strengths:

The authors perform a comprehensive analysis of FipA, including phenotyping of mutants, protein localization, localization dependence, and domains of FipA necessary for each. Moreover, they perform a time-series analysis indicating that FipA localizes to the cell pole likely prior to, or at least coincident with, flagellar assembly. They also show that the role of FipA appears to differ between organisms in detail but the overarching idea that it is a flagellar assembly/localization factor remains convincing.

Weaknesses:

For me the comparative analysis in the different organism was on balance, a weakness. By mixing the data for each of the organisms together, I found it difficult to read, and take away key points from the results. In its current form, the individual details seem to crowd out the model.

---

## [Referee Report · Reviewer #2 (Public review)]

Summary:

The authors identify a novel protein, FipA, which facilitates recruitment of FlhF to the membrane at the cell pole together with the known recruitment factor HupB. This finding is key to understanding the mechanism of polar localization. By comparing the role of FipA in polar flagellum assembly in three different species from Vibrio, Shewanella and Pseudomonas, they discover that, while FipA is required in all three systems, evolution has brought different nuances that open avenues for further discoveries.

Strengths:

The discovery of a novel factor for polar flagellum development. A significant contribution to our understanding of flagellar evolution. The solid nature and flow of the experimental work.

Weaknesses:

All my concerns have been addressed. I find no weaknesses. A nice, solid piece of work.

---

## [Referee Report · Reviewer #3 (Public review)]

Summary:

The authors investigate how polar flagellation is achieved in gamma-proteobacteria. By probing for proteins that interact with the known flagellar placement factor FlhF, they uncover a new regulator (FipA) for flagellar assembly and polar positioning in three flagellated gamma-proteobacteria. They convincingly demonstrate that FipA interacts genetically and biochemically with previously known spatial regulators HubP and FlhF. FipA is a membrane protein with a cytoplasmic DUF2802 and it co-localizes to the flagellated pole with HubP and FlhF. The DUF2802 mediates the interaction between FipA and FlhF and this interaction is required for FipA function. FipA localization depends on HubP and FlhF.

Strengths:

The work is throughly executed, relying on bacterial genetics, cell biology and protein interaction studies. The analysis is deep, beginning with the discovery af a new and conserved factor, to the molecular dissection of the protein and probing localisation and interaction determinants. Finally, they show that these determinants are important for function and they perform these studies in parallel in three model systems.

Weaknesses:

Because some of the phenotypes and localisation dependencies differ somewhat between model systems, the comparison is challenging to the reader because it is sometimes not obvious what these differences mean and why they arise.

---

## [Author Response]

The following is the authors’ response to the original reviews.

**eLife assessment**
This important research uses an elegant combination of protein-protein biochemistry, genetics, and microscopy to demonstrate that the novel bacterial protein FipA is required for polar flagella synthesis and binds to FlhF in multiple bacterial species. This manuscript is convincing, providing evidence for the early stages of flagellar synthesis at a cell pole; however, the protein biochemistry is incomplete and would benefit from additional rigorous experiments. This paper could be of significant interest to microbiologists studying bacterial motility, appendages, and cellular biology.

We are very grateful for the very positive and helpful evaluation.

**Joint Public Review:**
Bacteria exhibit species-specific numbers and localization patterns of flagella. How specificity in number and pattern is achieved in Gamma-proteobacteria needs to be better understood but often depends on a soluble GTPase called FlhF. Here, the authors take an unbiased protein-pulldown approach with FlhF, resulting in identifying the protein FipA in *V. parahaemolyticus*. They convincingly demonstrate that FipA interacts genetically and biochemically with previously known spatial regulators HubP and FlhF. FipA is a membrane protein with a cytoplasmic DUF2802; it co-localizes to the flagellated pole with HubP and FlhF. The DUF2802 mediates the interaction between FipA and FlhF, and this interaction is required for FipA function. Altogether, the authors show that FipA likely facilitates the recruitment of FlhF to the membrane at the cell pole together with the known recruitment factor HupB. This finding is crucial in understanding the mechanism of polar localization. The authors show that FipA co-occurs with FlhF in the genomes of bacteria with polarly-localized flagella and study the role of FipA in three of these organisms: V. parahaemolyticus, S. purtefaciens, and P. putida. In each case, they show that FipA contributes to FlhF polar localization, flagellar assembly, flagellar patterning, and motility, though the details differ among the species. By comparing the role of FipA in polar flagellum assembly in three different species, they discover that, while FipA is required in all three systems, evolution has brought different nuances that open avenues for further discoveries.Strengths:The discovery of a novel factor for polar flagellum development. The solid nature and flow of the experimental work.The authors perform a comprehensive analysis of FipA, including phenotyping of mutants, protein localization, localization dependence, and domains of FipA necessary for each. Moreover, they perform a time-series analysis indicating that FipA localizes to the cell pole likely before, or at least coincident with, flagellar assembly. They also show that the role of FipA appears to differ between organisms in detail, but the overarching idea that it is a flagellar assembly/localization factor remains convincing.The work is well-executed, relying on bacterial genetics, cell biology, and protein interaction studies. The analysis is deep, beginning with discovering a new and conserved factor, then the molecular dissection of the protein, and finally, probing localization and interaction determinants. Finally, the authors show that these determinants are important for function; they perform these studies in parallel in three model systems.Weaknesses:The comparative analysis in the different organisms was on balance, a weakness. Mixing the data for the organisms together made the text difficult to read and took away key points from the results. The individual details crowded out the model in its current form. Indeed, because some of the phenotypes and localization dependencies differ between model systems, the comparison is challenging to the reader. The authors could more clearly state what these differences mean, why they arise, and (in the discussion) how they might relate to the organism's lifestyle.More experiments would be needed to fully analyze the effects of interacting proteins on individual protein stability; this absence slightly detracted from the conclusions.

We have tried our best to improve the manuscript according to the insightful suggestions of the reviewers. Please find our answers to the raised issues below.

**Reviewer #1 (Recommendations For The Authors):**

We are very grateful to this reviewer for the very positive evaluation and the great suggestions to improve the manuscript.

I think there is value to the comparative analysis but how to present it in such a way that the key similarities and differences stand out is the challenge. Perhaps a table that compares the three datasets is sufficient. Or tell the story of *V. parahaemolyticus* first to establish the model, followed by comparative analysis of the other two organisms highlighting differences and relegating similarities to supplemental?

We agree that the our previous presentation of our comparative analysis made it very hard to follow the major findings and the general role(s) of FipA, and we are very grateful for the suggestions on how to improve this. We have decided to change the presentation as the reviewer recommended. We used *V. parahaemolyticus* as a ‚lead model‘ to describe the role of FipA, and we then compared the major findings to the other two species. We hope that the story is now easier to follow.

This is not something that needs to be addressed in the text but I wanted to bring the protein SwrB to the authors' attention which may further expand FipA relevance. *Bacillus subtilis* uses FlhFG to somehow pattern flagella in a peritrichous arrangement and there are a number of striking similarities, in my opinion, between FipA and SwrB. The two proteins have very similar domain architecture/topology, both proteins promote flagellar assembly, and the genetic neighborhood/operon organization is uncannily similar. There are other more minor similarities dependent on the organism in this paper.Phillips, Kearns. 2021. Molecular and cell biological analysis of SwrB in *Bacillus subtilis*. J Bacteriol 203:e0022721Phillips, Kearns. 2015. Functional activation of the flagellar type III secretion export apparatus. PLoS Genet 11:e1005443.

We thank this reviewer for pointing out these intriguing similarities. For this study we have decided to exclusively concentrate on polarly flagellated bacteria. FlhF und FlhG are also present in *B. subtilis* where they play a role in organizing flagellation, but we feel that this would be out of scope for this manuscript.

**Reviewer #2 (Recommendations For The Authors):**

We would like to thank this reviewer for the very positive evaluation and for pointing out several issues to strengthen the story.

Figure 3A data are problematic since everything is too small to visualize. Since these are functional GFP fusions (or mCherry for 2E data), why are they not presented in color?Again - why are color figures not used to help the reader in Fig 4A and 5F & 5G to confirm what is asserted?Again, it is difficult to see the images presented. It is asserted that FipA is recruited to the cell pole after cell division and before flagellum assembly, but one has to take their word for it.

We fully agree that in some case the localization pattern is hard to see on the micrographs presented. We have, therefore, provided enlarged micrographs in the supplemental part which allow to better see the fluorescent foci within the cells. With respect to presentations in color – we found that this did not improve the visibility of localizations and therefore have decided to use the grayscale images.

Here, what is missing are turnover assays. Do FipA, FlhF, and HubP all co-localize as complex or is the absence of one leading to the protein turnover of other partners? I think this needs to be sorted out before final conclusions can be made.

Thanks for pointing out this important point. We have now provided western analysis which demonstrate that FipA and FlhF are produced and stable in the absence of the other partners (see Supplemental Figure 5). Stability of HubP as a general polar marker not only required for flagellation was not determined.

Minor comments:Line 58: change "around" to "in timing with"Line 79: what "signal" is transferred from the C-ring to the MS-ring. Are they not fully connected such that rotation is the entire structure - C-ring-MS-ring-Rod-Hook-Filament. Is it not the change in the relationship to the stator complex where the signal is transferred?Line 85: change "counting" to "control of flagellar numbers per cell"Line 110: change "is (co-)responsible for recruiting" to "facilitates recruitment of"

Thanks for pointing this out. We have adjusted the wording according to the reviewer’s suggestions.

Given that motility phenotypes vary on individual plates (volumes and dryness vary), why in Figure 2C are the motility assays for fipA and flhF mutants of *P. putida* done on different plates?

For better visualisation, we have rearranged the spreading halos for the figure. All strain spreading comparisons on soft agar were always conducted on the same plate due to the reasons this reviewer mentioned.

**Reviewer #3 (Recommendations For The Authors):**

We thank this reviewer for the very positive evalution and the great suggestions.

One possibility is to describe first all the results relating to FipA in Vibrio and then add the result sections at the end to illustrate the differences between Vibrio and Shewanella, and then Vibrio and Pseudomonas. This may make it easier to follow for the reader.

We agree that the our previous presentation of our comparative analysis made it very hard to follow the major findings and the general role(s) of FipA, and we are very grateful for the suggestions on how to improve this. We have decided to change the presentation as the reviewer recommended. We used *V. parahaemolyticus* as a ‚lead model‘ to describe the role of FipA, and we then compared the major findings to the other two species. We hope that the story is now easier to follow.

I would have liked to see some TEM analysis of flagella in *fipA*/*hubP* double mutants strains and was also wondering if FipA/FlhF/HubP colocalization had been studied in *E. coli* when all proteins are expressed together, at least with two bearing fluorescent tags.

Thanks for these great suggestions. In this study, we have concentrated on the localization of FlhF by FipA and HubP. HubP has multiple functions in the cell and may also affect flagellar synthesis to some extent in a species-specific fashion. Therefore, any findings would have to be discussed very carefully, so we have decided to leave that out for the time being.

With respect to the FipA/HubP/FlhF production in a heterologous host such as *E. coli*, this has been partly done (without FipA) in a second parallel story (see reference to Dornes et al (2024) in this manuscript). Rebuilding larger parts of the system in a heterologous host is currently done in an independent study. Therefore, we have decided not to include this already here.

**From the Reviewing Editor:**

We are grateful for handling the fair reviewing process, for the positive evaluation and the helpful hints.

The microscopy was inconsistent (DIC versus phase) for unclear reasons. Did using different microscopes impact the ability to acquire low-intensity fluorescence signals? Please add a sentence in the Methods section to clarify.

We are sorry for this inconsistency. As the imaging was carried out by different labs (to some part before the projects were joined), the corresponding preferred microscopy settings were used. We have added an explaining sentence to the Methods section.

Also, some subcellular fluorescence localizations were not visible in the selected images (e.g., Figures 3 and 5). The reader had to rely on the authors' statements and analyses. The conclusions could be more robust with fluorescence measurements across the cell body for a subset of cells. The authors could provide this data analysis in the Supplemental; this measurement would more clearly show an accumulation of fluorescence at the cell pole, particularly in low-intensity images.

We fully agree that in some case the localization pattern is hard to see on the micrographs presented. Unfortunately, often the signal is not sufficiently strong to provied proper demographs. We have, therefore, provided enlarged micrographs in the supplemental part, which allow to better see the fluorescent foci within the cells.